# Feedback regulation of crystal growth by buffering monomer concentration

Samuel W. Schaffter [1], Dominic Scalise[1], Terence M. Murphy[2], Anusha Patel[1] & Rebecca Schulman [1,3,4 ✉]

Crystallization is a ubiquitous means of self-assembly that can organize matter over length scales orders of magnitude larger than those of the monomer units. Yet crystallization is notoriously difficult to control because it is exquisitely sensitive to monomer concentration, which changes as monomers are depleted during growth. Living cells control crystallization using chemical reaction networks that offset depletion by synthesizing or activating monomers to regulate monomer concentration, stabilizing growth conditions even as depletion rates change, and thus reliably yielding desired products. Using DNA nanotubes as a model system, here we show that coupling a generic reversible bimolecular monomer buffering reaction to a crystallization process leads to reliable growth of large, uniformly sized crystals even when crystal growth rates change over time. Buffering could be applied broadly as a simple means to regulate and sustain batch crystallization and could facilitate the self-assembly of complex, hierarchical synthetic structures.

[1] Chemical & Biomolecular Engineering, Johns Hopkins University, Baltimore, MD 21218, USA. [2] Our Lady of Lourdes High School, Poughkeepsie, NY 12603, USA. [3] Department of Computer Science, Johns Hopkins University, Baltimore, MD 21218, USA. [4] Department of Chemistry, Johns Hopkins University, Baltimore, MD 21218, USA. ✉email: rschulm3@jhu.edu

Crystallization is a ubiquitous process that can create large-scale order from atomic or molecular components[1]. Crystal growth is critical in applications such as protein[2] and molecular structure determination[3], the manufacture of nanoparticles[4], catalysts[5], and photonic materials[6,7] and for the self-assembly of large-scale ordered materials from molecular components by design[8–14]. Living systems also form many ordered structures, such as calcium carbonate structures or cytoskeletal filaments, through crystallization[15–17]. However, controlling what products result from crystallization: crystals' structure, size, and quality is notoriously difficult because crystallization depends sensitively on monomer concentrations and physical parameters that determine the energetics of monomer attachment and other growth reactions[1,18].

The difference between the free monomer concentration and the critical monomer concentration—the free monomer concentration at which no net crystal growth occurs[19,20], defined as supersaturation, determines the chemical potential for crystallization, and thus the rate and extent of crystal growth[18,21]. At high supersaturation, new crystals can spontaneously nucleate. Frequent nucleation throughout a growth process leads to crystals with a range of crystal sizes. High supersaturation also causes fast monomer attachment, which can kinetically trap crystal defects[18,21–23] (Fig. 1a). Growth of uniformly sized crystals with few defects occurs only within a narrow regime of supersaturation (Fig. 1b), whose boundaries are strongly dependent on growth conditions[1]. Seed crystals are used to facilitate growth in this regime, as spontaneous nucleation is rare[23–25]. As monomers are depleted as crystals grow, supersaturation must be maintained to grow large crystals either by using continuous flow reactors that provide fresh monomers to maintain a constant chemical potential during growth[26,27] or by using protocols such as continuous temperature decrease that lower the critical monomer concentration as monomers are depleted[1,22,28]. These methods must be developed and optimized for each specific crystallization process, as the physics of crystal growth depends on the types of monomers and seeds used and their concentrations[25]. These methods can require sophisticated equipment, fine tuning of reaction parameters, and are restricted in their applicability. Flow reactors, for example, cannot be used to control crystallization in confined environments such as micelles or living cells[9].

While engineers have generally used physical means of regulating crystallization, cells often use chemical reactions to regulate monomer concentrations during crystal growth[29,30]. Such reactions control tubulin turnover and availability during microtubule growth, for example[31]. These regulating chemical reactions provide closed loop feedback, allowing a cell to adaptively regulate growth in response to changes in the rate of monomer depletion, i.e., system load. For example, the regulation of active tubulin concentrations sustains microtubule growth even as the number of active microtubule organizing centers rapidly increases during cell division or migration[20,32]. This ability to continuously regulate the chemical potential for growth through varying growth conditions and loads is critical to cells' capabilities to build complex hierarchical and dynamic structures through crystallization[33,34]. Implementing chemical feedback regulation in synthetic crystallization processes (Fig. 1c) could thus make it possible to achieve similarly robust growth during complex hierarchical assembly processes with time-varying loads.

Cells typically use complex, precisely tuned reaction networks to regulate monomer concentrations during growth and similar types of synthetic biochemical networks have been proposed to regulate biomolecular concentration in vitro[35,36]. To regulate crystal growth using a simpler mechanism, we propose a single reversible, bimolecular reaction that effectively maintains the monomer concentration within a narrow regime of low

supersaturation as monomers are depleted during growth, thus making it possible to reliably grow large, uniformly sized crystals (Fig. 1c). Such a reversible reaction can act analogously to a pH buffer, where a reversible reaction between a weak acid and its conjugate base resists changes to hydrogen ion concentration, to buffer monomer concentration[37]. The equilibrium concentration of the reversible reaction defines the monomer's setpoint concentration and as monomers are depleted during growth, Le Chatelier's principle counteracts monomer depletion by resisting the disturbance to equilibrium. This buffering reaction can thus create a feedback loop that resists changes to the monomer concentration setpoint. The ubiquity of the buffering reaction suggests how it could be applied to a wide variety of crystallization processes under a range of physical conditions.

To demonstrate how monomer buffering can regulate crystallization, we grow DNA nanotubes from oligomeric DNA monomers[8,38] while buffering the nanotube monomer concentration. DNA nanotube growth is a well-understood model system for studying crystallization[8,23,39,40]. Watson–Crick hybridization of single-stranded overhangs, or sticky ends, drives the nanotube growth process (Fig. 2a) and a cylindrical DNA origami seed that mimics the nanotube growth face can be used to specifically control when and where nanotube growth

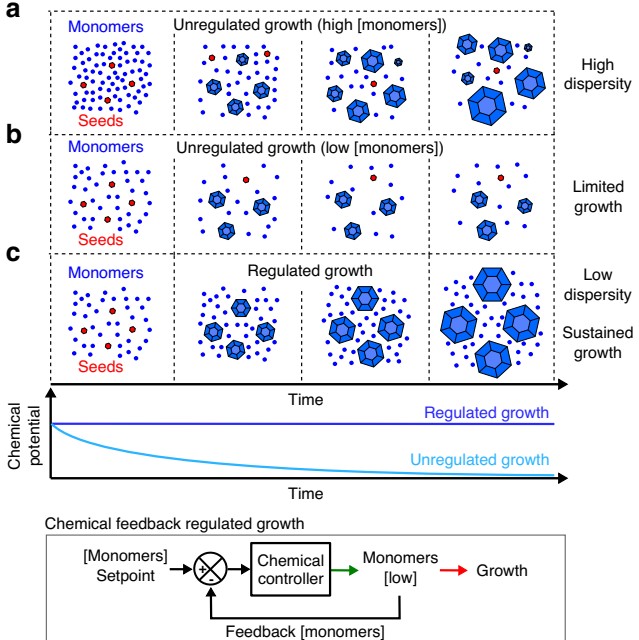

**Fig. 1 Crystal size, size dispersity, and quality are shaped by growth conditions, such as the initial monomer concentration and how fast monomers are depleted during crystallization.** Schematics show 3D crystals (blue) growing in the presence of seeds (red). **a** Far above the critical monomer concentration, new crystals homogeneously nucleate (i.e., not from seeds) continuously while other crystals grow. As a result, the crystals that form exhibit high size dispersity. Under these conditions, monomer addition to crystals is strongly forward biased, which means that crystals retain defects that form during growth. **b** When crystals are grown at a monomer concentration just above the critical monomer concentration, no homogenous nucleation occurs and crystals grow uniformly from seeds. However, crystals remain small because the monomer concentration rapidly reaches the critical concentration, halting growth. **c** When monomer concentration is regulated by a chemical feedback loop that holds the monomer concentration just above the critical concentration even as monomers are depleted by crystal growth, sustained growth of large, uniformly disperse crystals can be achieved.

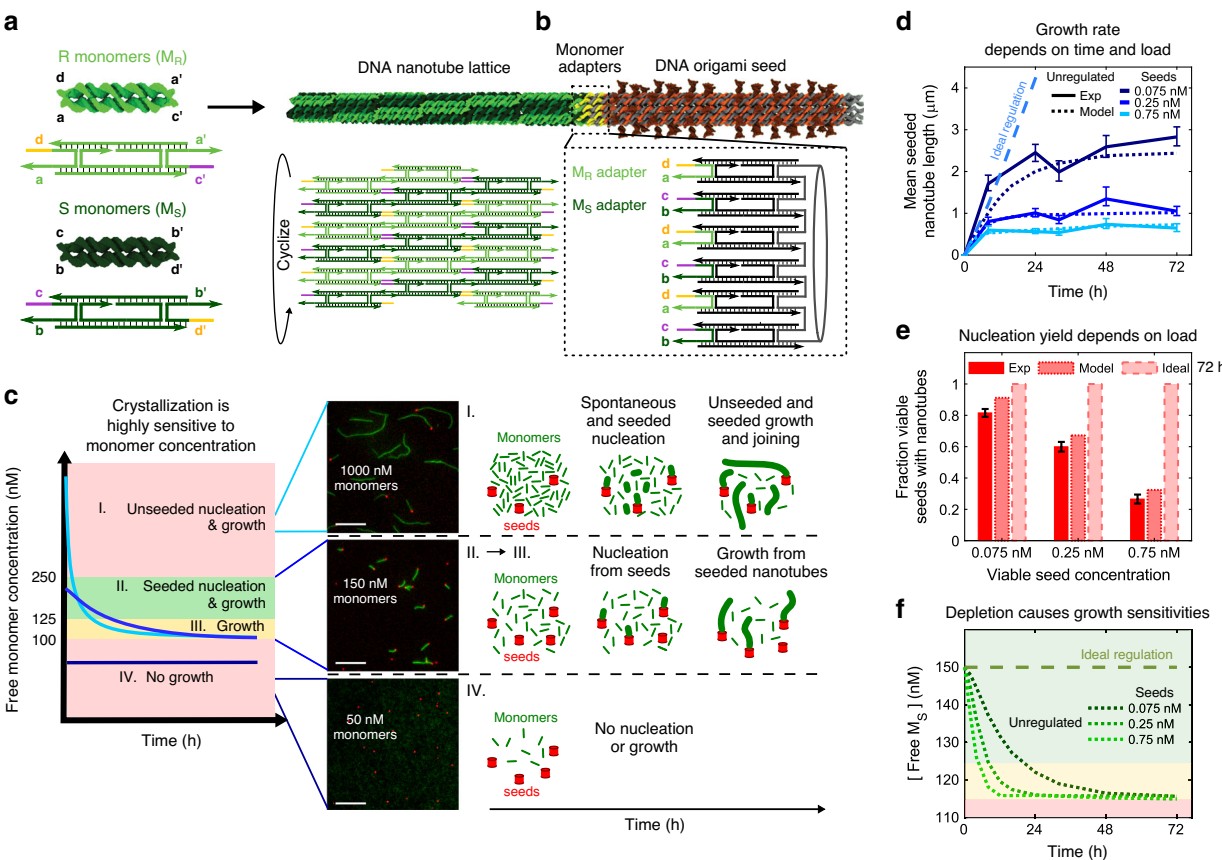

**Fig. 2 DNA nanotubes and unregulated DNA nanotube growth. a** Left: DNA monomers are composed of five DNA strands folded into rigid double crossover structures[64]. Right: Two monomer types co-assemble to form a cylindrical lattice via Watson–Crick hybridization of monomer sticky ends. **b** A DNA origami seed that presents monomer sticky ends at one edge (inset) acts as a stable nucleus from which nanotubes can grow without a significant energy barrier to nucleation. **c** At high monomer concentrations, spontaneous nanotube nucleation, growth, and joining all occur (regime *I.*). At intermediate monomer concentrations, DNA nanotubes nucleate and grow from seeds but spontaneous nucleation is rare (regime *II.*). The presence of a small energy barrier to nucleation from the seeds[23] results in a regime where growth from existing nanotubes is favorable but nucleation from additional seeds is rare (regime *III.*). At monomer concentrations below the critical concentration, no nanotube growth occurs (regime *IV.*). Fluorescence micrographs depict nanotubes (green) and seeds (red) after 24 h of growth at different initial monomer concentrations. Scale bars 10 µm. Supplementary Note 6 describes how the cutoffs for growth regimes were determined. **d–f** Results of experimental and simulated nanotube growth with 150 nM monomers at different seed concentrations (see "Methods" and Supplementary Note 4). **d** Mean lengths of seeded nanotubes measured in experiments (solid lines) and simulations (dashed lines). Error bars represent 95% confidence intervals from bootstrapping. **e** Fractions of viable seeds (Supplementary Note 7) that nucleated nanotubes after 72 h. Error bars on proportions represent 95% confidence intervals. The sample sizes (at least 50 nanotubes and seeds) for every timepoint of each sample are tabulated in Supplementary Note 14. **f** Free S monomer concentrations during simulations of growth. Shaded regions correspond to the growth regimes in **c**. Ideal regulation results in **d–f** are from simulations of nanotube growth without monomer depletion.

occurs[23,38,41] (Fig. 2b). However, there is a narrow range of monomer concentrations where DNA nanotubes grow only from seeds[42] (Fig. 2c).

In this work, we buffer DNA nanotube monomer concentrations during growth using a DNA strand displacement reaction[37] that facilitates the reversible exchange of active and inactive monomers. Through both simulations and experiments we demonstrate that regulating nanotube growth with monomer buffering reduces monomer depletion effects by maintaining the monomer concentration within a seeded nucleation and growth regime during crystallization. By mitigating monomer depletion, buffering allows more than an order of magnitude more monomers to be incorporated into nanotubes in a low supersaturation growth process than is possible in an unregulated growth process, facilitating the growth of large crystals with low-length dispersity. We also show how regulated growth, through feedback, adapts to changes in load over time, demonstrating how buffering might enable controlled dynamic or hierarchical growth processes to occur reliably. This work introduces a paradigm for regulating

crystallization through chemical feedback that controls the flow of chemical potential during growth and the simplicity of the chemical buffering mechanism suggests it should be generalizable to a range of chemical processes.

## Results

**DNA nanotube crystallization is highly sensitive to monomer concentration during growth.** How DNA nanotubes assemble in batch (i.e., closed vessel) reactions depends sensitively on the free monomer concentration and modes of assembly can be classified into four regimes where different nucleation and growth processes become important (Fig. 2c). At monomer concentrations just above the critical concentration, nanotubes grow only from seeds allowing control over when and where crystallization occurs.

Nanotubes assembled under these conditions are also monodisperse in width and have few defects because growth is slow and highly reversible[23]. However, there is only a narrow monomer

concentration regime where this type of seeded growth occurs. Above this regime, nanotubes primarily nucleate homogeneously rather than from seeds, and are disparate in size[8]. In the seeded growth regime, the higher the seed concentration (i.e., the load on monomer supply), the faster monomers are depleted. The presence of a small nucleation barrier[23] to growth from seeds produces a sub-regime (regime III) where nucleation from seeds is rare, which further complicates the growth process (Fig. 2c).

To understand how seeded nanotube growth rates change with time and load, we grew nanotubes from 150 nM monomers and either 0.01, 0.33 or 1 nM seeds and measured the lengths of nanotubes and the fractions of seeds that had nucleated nanotubes during each process after different times using fluorescence microscopy ("Methods"). We used these measurements to find rate constants of monomer attachment and detachment and the nucleation barrier to growth from the seed[23] that best recapitulated our results in a stochastic kinetic model of nanotube growth[23] (Supplementary Note 4). We found that only about 75% of the seeds could nucleate nanotubes, even when supersaturation was high (Supplementary Note 6), possibly because some seeds have structural defects, which make them nonviable. We, therefore, present all experimental and simulation data with respect to the concentration of viable seeds (0.075, 0.25, 0.75 nM) rather than the total concentration of seeds added (0.01, 0.33, 1 nM). In both experiments and simulations, nanotubes initially grew quickly, then slowed and stopped, consistent with shifts in monomer concentration from regimes II to IV. Higher seed concentrations (i.e., higher loads) depleted monomers faster, so that growth stopped sooner and nanotubes reached shorter final lengths (Fig. 2d, f). At high loads monomer depletion into regime III occurred so quickly that not all seeds had time to nucleate, causing the fraction of seeds that nucleated nanotubes to depend on load (Fig. 2e, f).

Monomer depletion during unregulated growth thus changes the rates and type of growth that occurs in ways that are complex to predict and control. Choosing a final length by choosing growth time, a typical method for controlling a crystallization process, would be difficult because growth occurs at a nonlinear rate. Changing the concentration of seeds in principle could allow the dynamics of the reaction to be tuned, but changing this concentration also changes the number of nanotubes, the fraction of seeds that nucleate, i.e., the yield, and the final lengths that nanotubes reach. In principle, increasing monomer concentrations could combat depletion effects, but at high monomer concentrations, seeds are not required for growth (Supplementary Fig. 8) and the majority of nanotubes nucleate on their own even when seeds are present (Supplementary Fig. 9). Thus, control over where and how many nanotubes grow is lost. Unseeded growth also results in much broader distributions of nanotube lengths than seeded growth (Supplementary Fig. 9).

Simulations suggest the difficulties of controlling nanotube growth would be avoided if monomers were not depleted during growth: crystals should nucleate and grow at a constant rate irrespective of load (Fig. 2d–f, ideal), and seeds should all nucleate nanotubes even when the seed concentration is high. Using such a process, many low-defect crystals could be grown to precise final lengths. We thus asked whether we might use chemical reactions to maintain a specific monomer concentration setpoint during growth (Fig. 1c).

**Regulating nanotube growth by buffering monomer concentrations**. We sought to design chemical reactions that would convert inactive monomers into active monomers as needed to maintain a monomer concentration setpoint during growth, thereby acting as a feedback controller. A large initial supply of

inactive monomers would ensure that this mechanism could maintain this setpoint concentration even after substantial growth (Fig. 3a).

We hypothesized that we could use a single reversible bimolecular reaction to regulate monomer concentration, a simple reaction scheme that could be adopted to regulate many crystallization processes. This reaction would regulate monomer concentration via a mechanism analogous to that which regulates pH in a buffered solution[37]. It could maintain, or buffer, the concentration of a monomer at a specific, easily tunable setpoint (Fig. 3b). As there were two monomer types, we needed a buffering reaction for each monomer type. In this scheme, an inactive monomer ($I_i$) reacts reversibly with a Producer ($P_i$) complex to create an active monomer ($M_i$) and a Consumer ($C_i$) strand. $i$ refers to the type of monomer species, either R or S (Fig. 3b and Eq. (1)). We termed $I_i$, $P_i$, and $C_i$ the monomer buffering species.

$$I_i + P_i \underset{k_{r,i}}{\overset{k_{f,i}}{\rightleftharpoons}} C_i + M_i \tag{1}$$

The monomer concentration setpoint that the buffering reaction maintains is the equilibrium concentration of active monomers (Fig. 3c, top box and Eq. (2)):

$$[M_i]_{eq} = \frac{k_{f,i}}{k_{r,i}} \frac{[I_i]_{eq}[P_i]_{eq}}{[C_i]_{eq}} = K_{eq,i} \frac{[I_i]_{eq}[P_i]_{eq}}{[C_i]_{eq}} \tag{2}$$

As the reaction in Eq. (1) is in dynamic equilibrium, Le Chatelier's principle induces feedback control on monomer $i$'s concentration during crystal growth: as active monomers are depleted, the monomer buffering reaction will shift its balance rightwards, producing more active monomers to resist a change in the setpoint (equilibrium) monomer concentration[37] (Fig. 3c, bottom box). This change depletes $[I_i]$, $[P_i]$ and increases $[C_i]$, but if $[I_i]$, $[P_i]$, $[C_i] \gg [M_i]$ these changes will be proportionally small, so $[M_i]$ will return to ~$[M_i]_{eq}$. Over time, however, even for $[I_i]$, $[P_i]$, $[C_i] \gg [M_i]$ the buffer's capacity will be depleted. The buffer's capacity is defined as the amount of $[M_i]$ that can be produced before the setpoint concentration drops by a specified percentage (e.g., 10%); the capacity is defined for this specified percentage. The buffer's capacity is given by Eq. (3) and has units of molarity:

$$c^-\left([I]_o + [P]_o + [C]_o\right), \tag{3}$$

where $c^-$ is a unitless parameter, called the capacity coefficient, that is a function of the specified percentage drop in setpoint concentration, and also varies by a factor of about five over the span of initial ratios of the concentrations of the buffering species ($c^-$ is ~0.01 for a 10% setpoint drop with roughly equimolar buffering species concentrations). See ref. [37] for the derivation of and full expression for $c^-$.

To implement monomer buffering, we first designed inactive monomers[43,44] consisting of four of the five oligonucleotides that make up an active monomer; these structures lacked the two sticky ends that mediate monomer binding to a seed or the facet of a nanotube grown from a seed (Fig. 3d and Supplementary Fig. 12). We then designed Producer and Consumer species that could react with the inactive and active monomers via toehold-mediated DNA strand displacement reactions[37,45] to orchestrate reversible inactive/active monomer exchange (Fig. 3d). The use of toehold-mediated strand displacement make it possible to set the values of the forward and reverse buffering reaction rate constants ($k_f$ and $k_r$), and thus the approximate equilibrium constant and buffering response time, within an order of magnitude by selecting toeholds ($e/e'$ and $d/d'$ in Fig. 3d) with specific binding energies[45]. We selected forward and reverse

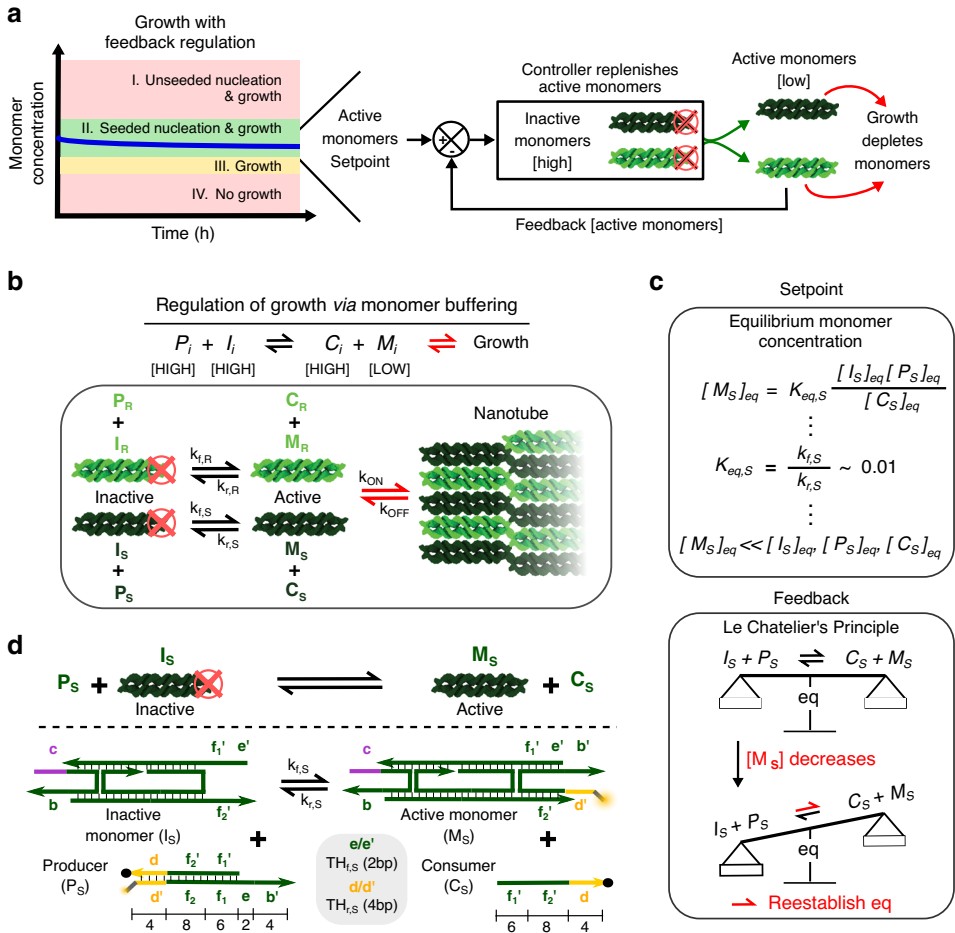

**Fig. 3 Regulating nanotube growth by buffering monomer concentrations. a** Schematic of a feedback controller for regulating growth. An ideal controller could maintain the monomer concentration within the seeded nucleation and growth regime by converting inactive monomers into active monomers as monomers are depleted by growth. **b** Overview of a scheme to regulate nanotube growth by buffering monomer concentrations via reversible conversion of monomers in inactive and active states. **c** Monomer buffering provides setpoint and feedback control. Top: The setpoint is determined by the equilibrium monomer concentration. A large concentration of I, P, and C can be used to regulate a low concentration of M. Bottom: Feedback is provided by Le Chatelier's principle. As monomer concentration decreases during growth, the monomer buffering reaction will produce more active monomers to rebalance the forward and reverse reaction rates, thereby resisting a change in equilibrium. **d** A DNA strand displacement reaction network for buffering S monomer concentrations. An inactive monomer complex ($I_S$) reacts with a Producer complex ($P_S$) via a strand displacement reaction initiated by a single-stranded toehold domain ($TH_{f, S}$) to produce an active monomer ($M_S$) and Consumer strand ($C_S$). $C_S$ can react with an active monomer via another toehold domain ($TH_{r, S}$) to reverse the active monomer production reaction. Numbers indicate domain lengths in bases. An analogous network was also designed for the R monomers (Supplementary Fig. 12).

toehold lengths of 2 and 4 bases, respectively, which correspond to predicted values of $k_{f,i}$ on order $1 \times 10^2\,\mathrm{M^{-1}\,s^{-1}}$ and $k_{r,i}$ on order $1 \times 10^4\,\mathrm{M^{-1}\,s^{-1}}$[45] and a $K_{\mathrm{eq},i}$ on order 0.01. These toehold lengths should ensure fast buffer response times[37] and a $K_{\mathrm{eq},i}$ around 0.01 allows micromolar concentrations of $I_i$, $P_i$, and $C_i$ on the same order. Keeping these concentrations on the same order maximizes the capacity coefficient ($c^-$) in Eq. (3) for a given total concentration of buffering species[37].

**Regulating growth with monomer buffering reduces growth sensitivity to load and time.** We first used simulations to quantify, for reasonable concentrations of the buffering species, how much monomer buffering could reduce the depletion of free monomers during nanotube growth. To obtain a high capacity, we selected concentrations of the inactive monomers ($I_i$) and $P_i$ to both be 5.5 µM (>35 times the setpoint concentration).

We then set $[C_i]_0 = 1.69\,\mu M$ to obtain an active monomer setpoint of 155 nM (Supplementary Note 8). We incorporated the monomer buffering reactions with the designed rate constants

into our stochastic kinetic model (Supplementary Note 9) and simulated nanotube growth for a range of seed concentrations.

These simulations showed that monomer buffering reduced, but did not eliminate, the decrease in active monomer concentration during growth (Fig. 4a): each active monomer created consumes an inactive monomer and $P_i$ complex and produces a $C_i$ strand; altering the ratio of these species concentrations during growth, which subsequently decreases the setpoint (Eq. (2) and Supplementary Fig. 14).

Despite not completely eliminating monomer depletion, simulations still predicted monomer buffering could maintain the active monomer concentration within the seeded nucleation and growth regime 3–8 times longer (depending on the seed concentration) than would be maintained during unregulated growth (Fig. 4a). The increased time in the seeded growth regime resulted in much more linear nanotube growth rates, especially at lower seed concentrations. Simulations also predicted nanotubes would grow much longer (Fig. 4b) while still maintaining low dispersity in their lengths (Supplementary Fig. 15). Finally, in

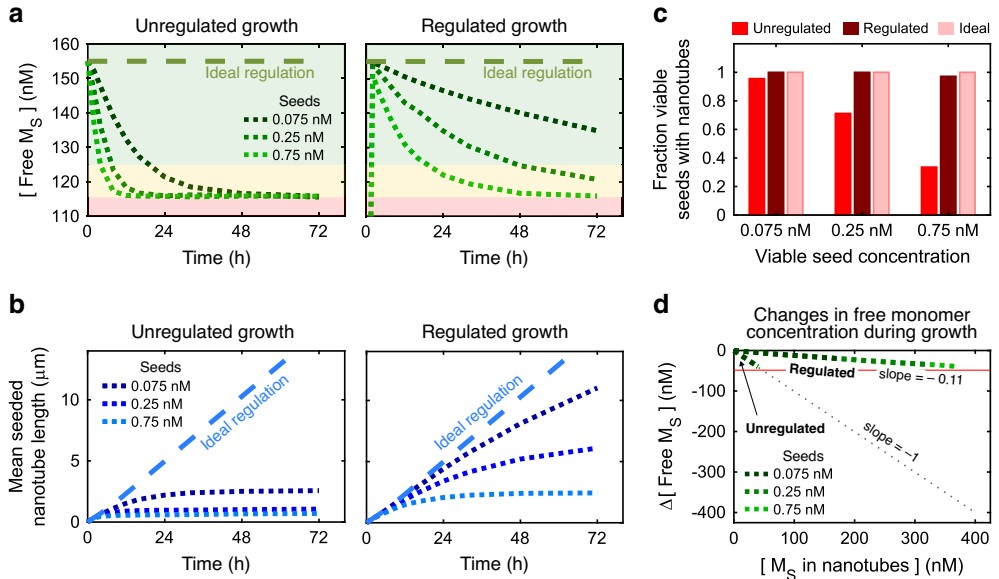

**Fig. 4 Effects of buffer-regulated growth predicted by kinetic simulations. a** Buffer-regulated growth reduces the rate at which the free monomer concentration decreases during a nanotube growth process. Shaded regions correspond to the growth regimes in Fig. 2c. The concentration of the S monomer is shown; depletion of the two monomer types should happen at the same rate. In regulated growth, the free $M_S$ concentration starts at zero and quickly equilibrates to the setpoint concentration. **b** Compared to unregulated growth with 150 nM monomers (left plots), nanotubes are predicted to grow much longer during buffer-regulated growth (right plots). **c** Predicted fractions of viable seeds with nanotubes after 72 h of buffer-regulated, unregulated and ideal (no depletion) growth. **d** Predicted changes in free $M_S$ concentration as a function of the total concentration of $M_S$ incorporated into nanotubes. Without regulation, free monomer concentration decreases at the same rate as monomer incorporation (slope = −1); less-negative slopes indicate resistance to monomer depletion. For ideal regulation, there would be no change in free monomer concentration (slope = 0). The green dashed lines show the change in $M_S$ vs. total monomer incorporation during buffer-regulated growth. These lines have the same slope for all seed concentrations (and therefore overlap) because the amount the setpoint drops per $M_S$ incorporated is a constant irrespective of load. The slope of these overlapping lines is the depletion ratio. The red line indicates the critical monomer concentration for growth. When the change in free $M_S$ reaches this line, growth will stop. The higher the seed concentration the faster this line will be reached and the buffer exhausted. This analysis indicates that buffer-regulated growth is predicted to incorporate nearly 10-fold more monomers into nanotubes than unregulated growth (roughly 360 nM vs. 40 nM, respectively). In unregulated growth simulations, the initial monomer concentrations were each 155 nM. Regulated growth simulations were conducted with $[I_i]_o = [P_i]_o = 5.5\,\mu M$ and $[C_i]_o = 1.69\,\mu M$ for both R and S monomers, resulting in a setpoint active monomer concentration of 155 nM. See Supplementary Note 9 for additional simulation details.

simulations of regulated growth, the monomer concentration was maintained within the seeded nucleation and growth regime long enough for all of the seeds to nucleate at all seed concentrations, producing maximum yields across all growth conditions (Fig. 4c).

To quantify how well the monomer buffering reactions should resist changes to the monomer concentration setpoint during growth, we used the simulation results to compare how much the free active monomer concentration changed relative to the total concentration of active monomers incorporated into nanotubes. We termed the ratio of these two quantities the depletion ratio as it is a measure of how well the buffer resists monomer depletion. In an unregulated growth process the depletion ratio is 1:1; each nanomolar of monomer incorporated into nanotubes corresponds to a nanomolar drop in the free monomer concentration. In simulations of buffer-regulated growth, this ratio was nearly 1:10; the free monomer concentration only drops by 1 nM for every 10 nM of monomers incorporated into nanotubes. Although the seed concentration (i.e., the load) dictates the rate at which monomers are depleted (and the total amount of monomers depleted during a given growth time), the depletion ratio, i.e., the amount the setpoint concentration drops with respect to the concentration of monomers incorporated during growth, is the same irrespective of load (Fig. 4d). The depletion ratio is analogous to buffer capacity and is thus set by the initial concentration of the buffering species (Eq. (3)).

The total quantity of monomers that can be incorporated into nanotubes before the buffer is exhausted is the difference between the initial setpoint concentration and the critical concentration divided by the depletion ratio. Thus, the simulations predict that roughly 10-fold more monomers are incorporated into nanotubes for buffer-regulated growth than for unregulated growth before the setpoint monomer concentration drops to the critical concentration (Fig. 4d).

We next sought to test the predictions of the model experimentally. We used 5.5 μM each as initial inactive monomer ($I_i$) and $P_i$ concentrations, the value used in our simulations of buffer-regulated growth. The initial concentrations of $C_i$ that would achieve the desired active monomer setpoints of 155 nM depend on the precise strand displacement rate constants $k_{f,i}$ and $k_{r,i}$ (Supplementary Note 8). The designed toehold lengths suggested the mean expected rates but not the exact ones, so to determine the exact $C_i$ concentrations to use we characterized buffer-regulated growth using four different $C_i$ concentrations ranging from 1 to 1.69 μM and looked for conditions where significant seeded nanotube growth but little unseeded growth was observed, which would suggest an initial monomer setpoint concentration in growth regime II. Buffer-regulated growth with $C_i = 1.25\,\mu M$ resulted in more seeded nanotube growth than higher $C_i$ concentrations (Supplementary Fig. 16) and less growth without seeds than $C_i = 1\,\mu M$, with unseeded growth comparable to unregulated growth with 150 nM monomers (Supplementary Fig. 17). We thus selected $C_i = 1.25\,\mu M$ to further characterize buffer-regulated growth experimentally.

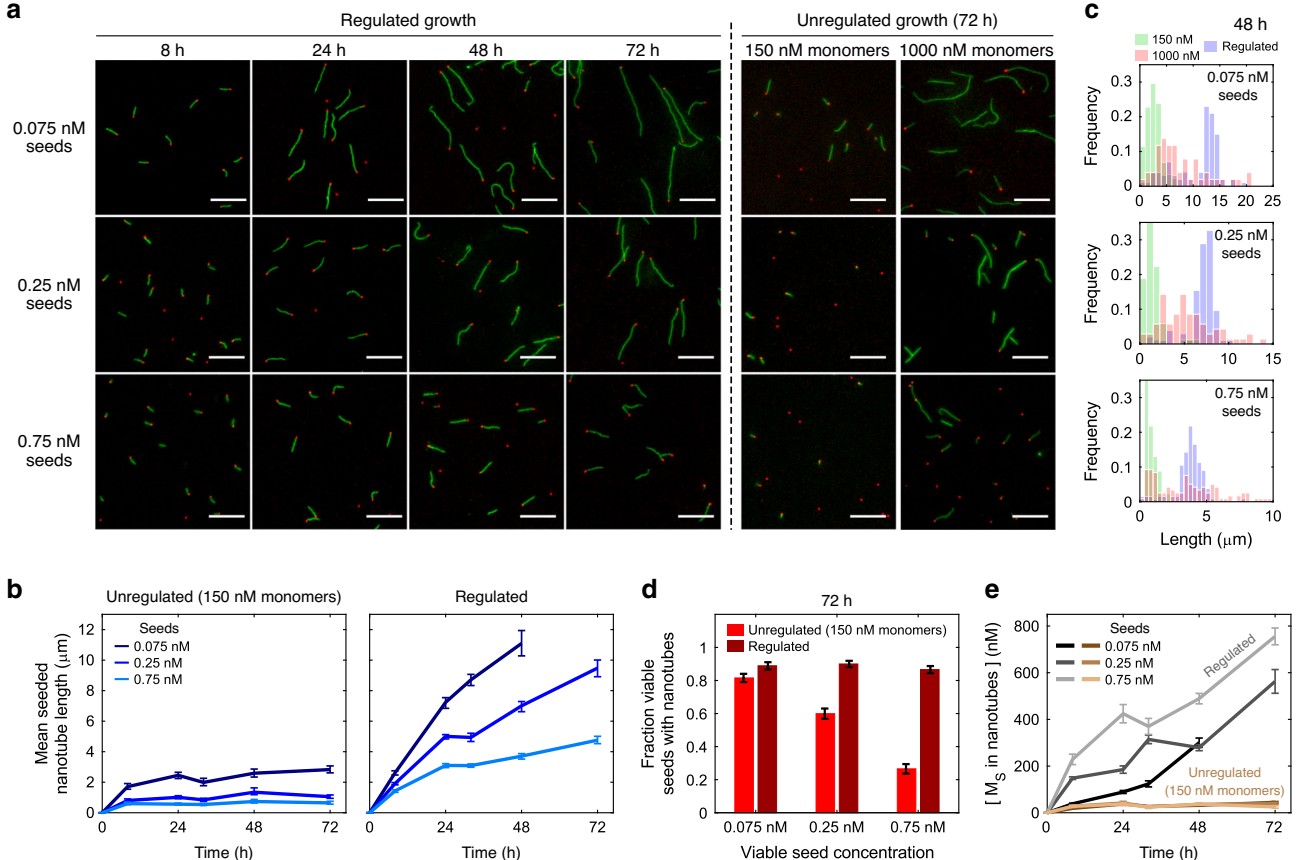

**Fig. 5 Comparing unregulated DNA nanotube growth with buffer-regulated growth. a** Fluorescence micrographs of DNA nanotubes after different durations of unregulated or buffer-regulated growth. Scale bars: 10 μm. **b** Mean lengths of seeded nanotubes during unregulated (left) or buffer-regulated growth (right) (Methods, Supplementary Fig. 18). Error bars represent 95% confidence intervals from bootstrapping. **c** Distributions of seeded nanotube lengths after 48 h from the experiments presented in **a**. For distributions of nanotube lengths at other timepoints, see Supplementary Fig. 19. **d** Fractions of viable seeds with nanotubes after 72 h of growth. Experimental variation (possibly due to pipetting) and sampling errors introduced slight variations in the fractions of viable seeds with nanotubes across timepoints; at most timepoints regulated growth resulted in a fraction near 1. The fractions for all timepoints are shown in Supplementary Fig. 10. Error bars on proportions represent 95% confidence intervals. **e** Mean concentrations of $M_S$ incorporated into nanotubes during growth. This value is roughly 35–40 nM for unregulated growth at all seed concentrations. Error bars represent standard deviation across images. Regulated growth experiments were conducted with $[I_i]_o = [P_i]_o = 5.5\,\mu M$ and $[C_i]_o = 1.25\,\mu M$ for both monomer types. The sample sizes for every timepoint of each sample are tabulated in Supplementary Note 14.

Under these conditions, nanotubes continued to grow over 72 h during buffer-regulated growth at rates at least a third of their initial growth rate. During unregulated growth with 150 nM monomers, nanotubes stop growing within 8 h. Thus, the nanotubes that formed in the buffer-regulated process were 5–10 times longer than those formed during unregulated growth (Fig. 5a, b). The nanotubes grown in the regulated process exhibited low-length dispersity (25% coefficient of variation (CV)) as expected for a controlled seeded growth process where all nanotubes nucleate from seeds at similar times. In contrast, unregulated growth with a high concentration of monomers (1000 nM) produced long nanotubes with high length dispersity (71% CV) (Fig. 5c) and high fractions of unseeded nanotubes (Supplementary Fig. 19), as expected for growth primarily via uncontrolled homogeneous nucleation and end-to-end joining[40]. Further, as predicted by our simulations, buffer-regulated growth resulted in nearly every seed nucleating a nanotube at all seed concentrations studied, in contrast to unregulated growth with 150 nM monomers, where higher loads depleted monomers so quickly that only a small fraction of seeds had time to nucleate (Fig. 5d). Together these results suggest that buffer-regulated growth sustains nanotube growth much longer than growth in the absence of regulation by keeping the concentration

of free active monomers within the seeded nucleation and growth regime.

We then quantified the concentration of S monomers, which were fluorescently labeled ("Methods"), incorporated into nanotubes during growth. At the highest seed concentration, we found after 72 h of buffer-regulated growth the total concentration of S monomers incorporated into nanotubes was roughly 18-fold higher than after the same duration of unregulated growth (Fig. 5e). Interestingly, nearly twice as many monomers were incorporated during buffer-regulated growth, nearly 750 nM, than simulations predicted (Fig. 4d). This suggests that the effective setpoint of the buffers characterized in experiments was slightly higher than the 155 nM setpoint used in our initial simulations. Simulations of buffer-regulated growth with $[C_i]_o$ set to the value used in experiments (1.25 μM) predicted a total amount of monomer incorporation close to the measured amount (Supplementary Note 10).

**Feedback regulation maintains growth within the seeded regime despite temporal changes in load.** A key characteristic of feedback regulation is the ability to adjust system output to accommodate changing loads, in the case of crystal growth, the

demand for free monomers. Such a capability to adapt to changes in load could make it possible to build hierarchical structures where growth sites change in number as they are sequentially activated and/or terminated during growth. For example, microtubule growth and branching is important during neuron development to build hierarchical axon networks[33].

To investigate how temporal load increases would influence nanotube crystallization, we designed an experiment in which the addition of more seeds after 24 h of growth would increase the load. Simulations suggested that after 24 h of unregulated growth with 150 nM monomers the monomer concentration had dropped out of the regime where nucleation from seeds was favorable, but for buffer-regulated growth, the monomer concentration was high enough that nanotubes could nucleate and grow from all of the added seeds (Supplementary Fig. 21).

For both buffer-regulated and unregulated growth, we grew nanotubes with S1 seeds for 24 h, then doubled the seed concentration by adding S2 seeds, which were identical to S1 except for their fluorescent label, and tracked growth for another 48 h (Fig. 6a). Consistent with our simulations, no growth was observed from the S2 seeds in unregulated growth while nearly all of both seed types nucleated and grew nanotubes for buffer-regulated growth (Fig. 6b–c).

Simulations also predicted that the rate of monomer production during regulated growth should increase when the seed concentration increases to compensate for the increased rate of active monomer depletion (Supplementary Fig. 21). We observed this increased rate of production in the buffer-regulated growth experiments: the sum of the mean lengths of nanotubes grown from the S1 and S2 seeds was 13.8 μm, meaning that >50 nM more S monomers were incorporated after 48 h (Fig. 6b) than when nanotubes grown were with the 0.075 nM seed concentration alone (mean length of 11.1 μm) (Fig. 5b).

A feedback regulation mechanism should also be able to decrease or cease output production if the load is decreased or eliminated. For example, if the growth load is eliminated, monomers should not accumulate to push the reaction into the unseeded regime as would be expected for open loop regulation. To test whether monomer buffering could compensate for the complete elimination of load, we added DNA origami caps, which attach to nanotube ends, preventing further growth, in excess of seeds after different growth times[38] (Fig. 6d). Caps halted nanotube growth (Fig. 6e) and importantly no significant unseeded growth was observed even long after growth ceased (Fig. 6f), indicating that net monomer production stopped when the load was eliminated.

**Buffer-regulated growth is not sensitive to reaction rates or competing reactions**. Our buffer-regulated growth experiments showed sustained growth of nanotubes and nucleation at all seeds, consistent with the predictions of simulations (Fig. 5), however, we observed that the initial rate of buffer-regulated growth was 30–40% slower in experiments than was predicted by our simulations (Fig. 7a). Given that more total monomers were incorporated into nanotubes experimentally than expected by our simulations (Figs. 5e and 4d, respectively), the slower growth rate was likely not the result of the monomer concentration setpoint being lower than expected. Competition with buffering species to bind to growth sites might also influence the growth rate by lowering the effective monomer attachment rate constant ($k_{ON}$), but simulations with reduced $k_{ON}$ values did not recapitulate the different initial growth rates we observed across seed concentrations (Supplementary Fig. 22).

We theorized this discrepancy arose instead because the rate constants for the monomer buffering reactions were lower than

expected, perhaps because the strand displacement processes involved the crossing from one helix to another, which imposes an energy barrier[46–48]. Indeed, simulations of regulated growth in which monomer buffering reaction rate constants were each two orders of magnitude lower than the values we initially assumed predicted growth kinetics that closely matched those measured experimentally (Fig. 7b and Supplementary Fig. 23). While the buffer that we designed should replenish monomers as quickly as they are depleted (Fig. 7c), a buffer with these lower rate constants cannot, with the result being that actual monomer concentration at a given time is lower than the monomer concentration setpoint. This lower monomer concentration results in slower growth and thus slower monomer depletion (Fig. 7d). However, despite the monomer buffering reactions likely being much slower than designed both fast and slow monomer buffering reaction networks maximize seed nucleation (Supplementary Figs. 22 and 23). Further, because both systems have similar chemical potentials, the predicted final lengths of nanotube are very similar (Supplementary Fig. 24). The fact that buffer-regulated growth is not sensitive to the exact values of the rates of the monomer buffering reactions might explain why monomer buffering works despite the fact that the two monomer types likely have different buffering reaction rates. It's likely that the setpoint concentration of one of the monomer types dictates the growth rate even if the concentration of the other monomer type is higher. Thus, it may be possible to regulate multi-component self-assembly processes by only buffering the concentration of one or a few components while supplying the rest of the components at high concentrations, assuming homogeneous nucleation is still suppressed.

Here, we regulated nanotube growth using an inactive monomer ($I_S$) concentration roughly 36 times that of the desired active monomer ($M_S$) concentration (5.5 μM and 150 nM, respectively). These conditions increased the total amount of monomers incorporated into nanotubes almost 20-fold compared to unregulated growth. Increasing the initial concentrations of the buffering species could increase the buffering capacity (Eq. 3) (Supplementary Note 12). However, increasing the concentrations of the buffering species would also increase the rates of unintended reactions (i.e., side reactions) involving the buffering species, which have complementary sequences with active monomers and nanotube growth facets. Such side reactions might slow down or prevent crystallization, potentially imposing an upper limit on the buffering capacity that could be achieved (Supplementary Note 13). For example, the $P_i$ complexes could bind to seeds or nanotube growth faces, effectively blocking growth; we observed that an unregulated growth process supplemented with increasing concentrations of $P_i$ complexes reduced the total amount of nanotube growth observed compared to unregulated growth without $P_i$ complexes (Supplementary Fig. 26). Active monomers could also transiently bind to inactive monomers, preventing these bound monomers from attaching to nanotubes. These transient interactions could thus lower the active monomer concentration during growth. We found active monomers incubated with increasing concentrations of only inactive monomers resulted in shorter nanotubes and fewer seeds nucleating nanotubes after 24 h than unregulated growth in the absence of inactive monomers. Surprisingly, we observed that no nanotube growth occurred at all in this unregulated growth process when inactive monomers were present at concentrations similar to those used in the buffer-regulated growth experiments in Fig. 5 (Supplementary Fig. 27). Regulated growth presumably still occurs because monomer buffering can partially compensate for the depletion of active monomers by transient interactions between the active and inactive monomers (Supplementary Fig. 27). Thus, feedback regulation could partially mitigate the

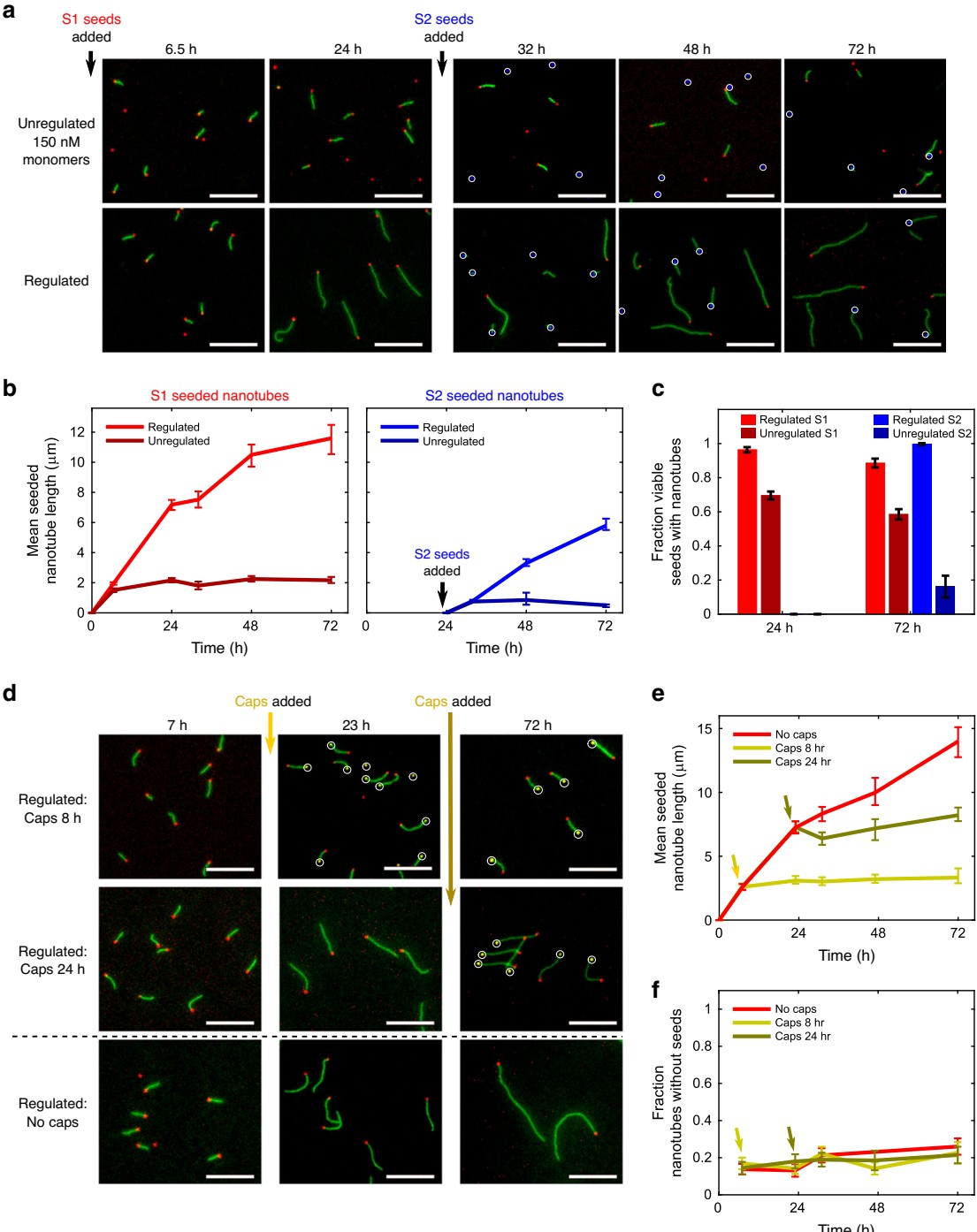

**Fig. 6 Feedback regulation maintains growth within the seeded growth regime even as load changes during growth. a** Fluorescence micrographs of seeded nanotubes during unregulated (top panel) or buffer-regulated (bottom panel) growth where 0.075 nM viable S1 seeds (red) were added at the beginning of the experiment and 0.075 nM viable S2 seeds (blue) were added after 24 h (increasing the load). S1 and S2 seeds differ only in their fluorescent labels. S2 seeds are circled in white for clarity. **b** Mean lengths of S1- and S2-seeded nanotubes during regulated and unregulated growth with a load increase. Error bars represent 95% confidence intervals from bootstrapping. **c** Fractions of S1 and S2 seeds with attached nanotubes during regulated and unregulated growth. Error bars on proportions represent 95% confidence intervals. **d** Fluorescence micrographs of seeded nanotubes during buffer-regulated growth with DNA origami caps added (0.2 nM) after either 8 (top panel) or 24 h (middle panel) to remove load. Caps are shown in yellow and circled in white for clarity. **e** Mean lengths of seeded nanotubes during buffer-regulated growth with and without load removal for the samples in **d**. Error bars represent 95% confidence intervals from bootstrapping. **f** Fractions of nanotubes not attached to seeds during buffer-regulated growth with and without the addition of caps. Error bars on proportions represent 95% confidence intervals. The 15–20% of nanotubes without seeds in all samples at all timepoints is likely the result of homogeneous nanotube nucleation. This level of background unseeded growth is consistent with other experiments for regulated and unregulated growth with 150 nM monomers (Supplementary Fig. 19). The sample sizes for every timepoint of each sample are tabulated in Supplementary Note 14. Unregulated growth reactions used 150 nM monomers. Regulated growth experiments were conducted with $[I_i]_o = [P_i]_o = 5.5\,\mu M$ and $[C_i]_o = 1.25\,\mu M$ for both monomer types. All scale bars: 10 μm.

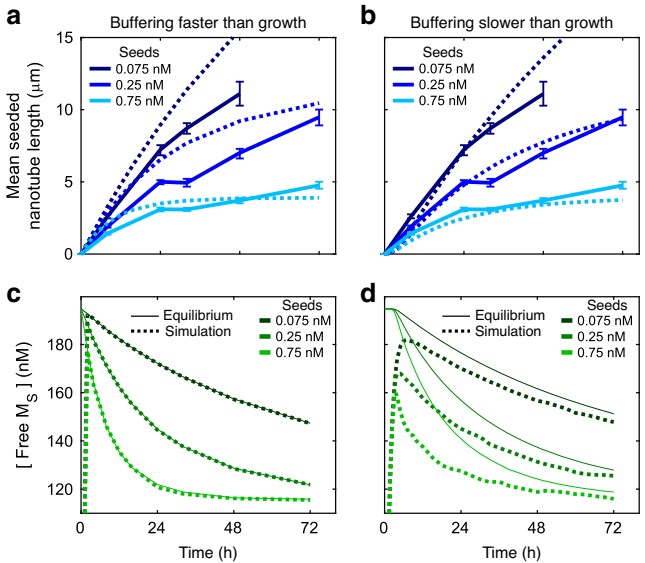

**Fig. 7 Monomer buffering appears to be slower than nanotube growth for our selected concentrations of monomer buffering species. a, b** Mean seeded nanotube lengths measured in experiments (solid lines) or simulations (dashed lines) during buffer-regulated growth. Error bars represent 95% confidence intervals from bootstrapping. **c, d** S monomer concentrations during simulations of buffer-regulated growth (dashed lines). Solid lines represent the theoretical equilibrium S monomer concentrations over the course of the experiments (Supplementary Note 8). Regulated growth simulations were conducted as described in Supplementary Note 9 with $[I_i]_o = [P_i]_o = 5.5 \,\mu M$ and $[C_i]_o = 1.25 \,\mu M$ for both monomer types. The forward and reverse buffering rate constants were $1 \times 10^2$ and $1 \times 10^4 \, M^{-1}s^{-1}$, respectively, for the simulations where monomer buffering was faster than growth (left panels) and $1 \times 10^0$ and $1 \times 10^2 \, M^{-1}s^{-1}$ for the simulations where monomer buffering was slower than growth (right panels). See Supplementary Note 11 for additional simulation details.

effects of side reactions during crystallization, even those imposed by the regulation mechanism itself. However, at concentrations >100 times that of the setpoint active monomer concentration, inactive monomer ($I_i$) and $P_i$ would significantly impede growth by blocking growth sites (Supplementary Note 13). Thus, in practice, designing an effective crystallization buffering scheme is a tradeoff between high capacity and detrimental side reactions; where the optimum concentrations of buffering species are likely 10–100 times that of the desired active monomer concentration.

## Discussion

Here, we have demonstrated a simple feedback mechanism based on chemical buffering[37] that resists changes in free monomer concentration, and thus chemical potential, during DNA nanotube crystallization. By buffering monomer concentrations, we were able to significantly extend nanotube growth times and maximize heterogeneous nucleation yields while producing crystals with low-length dispersity.

While chemical reactions have been used to control whether a structure forms[43,44,49,50], here we show how chemical reactions can also be used to ensure that structures form reliably and efficiently. In doing so, we have also shown that feedback to regulate chemical concentrations during assembly can be induced by a single, generic set of reversible reactions rather than networks for feedback requiring multiple, more complex reactions[35,36]. The simplicity of the mechanism for feedback

control used here suggests it should be applicable to a wide range of other self-assembly processes. For example, the DNA strand displacement mechanism could be modified to buffer crystallization of multi-dimensional DNA nanostructures[11,12,14,51–55], DNA-functionalized nanoparticles or colloids[10]. The buffering reaction is also simple enough that variants could conceivably be adopted to regulate the crystallization of proteins[2,9] or inorganic crystals[21] particularly for cases where inactive monomer precursors already exist and their conversion to active components can be controlled[56,57].

Physical methods such as continuous flow reactors[26] or auto-titration[58] could be used to control monomer concentrations during crystallization. We introduce a general means to implement similar control using chemistry in a closed system. Though mechanistically simple, the powerful feedback regulation imparted by monomer buffering should facilitate growth of complex hierarchical structures where growth sites are activated or deactivated during the crystallization process[38,59] or allow crystals to heal itself after damage by maintaining a constant growth potential[60]. This may be why processes such as actin and microtubule growth are so tightly regulated by coupled production and degradation reactions that also resist concentration changes.

While buffer-regulated crystallization dramatically increases the amount of nanotube growth that could occur in a batch reaction, it is not an ideal means of regulation: the setpoint monomer concentration continually decreases during growth (Fig. 4a). Although the rate of setpoint decrease can be reduced by raising concentrations of the monomer buffering species (Supplementary Note 12), higher concentrations can also increase the rates of undesired side reactions, ultimately limiting capacity (Supplementary Note 13). Extending or altering the buffering network that regulates crystallization could get around these limitations. For example, increased capacity might be achieved by buffering the concentrations of the species that buffer monomer concentration. Moreover, such chemical regulation might not only maintain a setpoint monomer concentration, but also orchestrate different regulatory programs that require up- or down- regulation of chemical potential during different stages of hierarchical self-assembly. Such networks might make the self-assembly of complex hierarchical devices and machines reliable and routine.

## Methods

**DNA components.** All oligonucleotides used in this study were synthesized by Integrated DNA Technologies (IDT). The sequences of the monomer buffering species are in Supplementary Note 1 M13mp18 DNA (7240 bases) was purchased from Bayou Biolabs (Cat# P-107). The sequences of the staple strands and the labeling strands for the DNA origami seed are the same as those used in previous studies[23,41] and are in Supplementary Note 2. The sequences of the adapter strands of the DNA origami seed are in Supplementary Note 2. The sequences for the DNA origami cap are in Supplementary Note 3. To prevent inactive monomers from creating a high background signal during nanotube imaging, the respective strands of the S producer complex ($P_S$) have a fluorophore and quencher such that the fluorescence is quenched; during monomer activation the strand with the fluorophore is incorporated into the active monomer and separated from the quencher so that the active monomer is fluorescent (Supplementary Note 1).

**Preparation of the monomer buffering species and DNA origami seeds and caps.** All DNA complexes and structures were prepared in an Eppendorf Mastercycler in 40 mM Tris-Acetate, 1 mM EDTA buffer supplemented with 12.5 mM magnesium acetate (TAEM). R and S inactive monomers were prepared separately with all of their strands present at 25 μM. $P_i$ complexes were prepared separately with each of their strands present at 50 μM. After mixing all relevant components, samples were thermally annealed[61,62] by first being heated to 90 °C for 5 min to ensure all strands were entirely single-stranded and then being cooled to 20 °C at −1 °C/min. Annealed complexes were stored at 4 °C until use.

The DNA origami seeds and caps were prepared as previously described[23,38,41]. The DNA origami seed is composed of a scaffold strand (M13mp18 DNA), 72 staple strands, and 24 adapter strands. The DNA origami cap is composed of a

scaffold strand (M13mp18 DNA) and 24 adapter strands. Seeds were labeled with fluorophores as previously described[41]. Seeds (except for S2 seeds) were labeled with atto488. S2 seeds and caps were labeled with atto647. The DNA origami seeds and caps were prepared in TAEM buffer with 5 nM M13mp18 DNA, 250 nM of each staple strand, 200 nM of each sticky end adapter strand (strand 4 or strand 2 for AS1-6 in Supplementary Notes 2 and 3, respectively) and 100 nM of all other adapter strands, 10 nM of each labeling strand, and 1000 nM of the fluorescently labeled strand. Biotinylated-BSA at a final concentration of 0.05 mg/mL (Cat# A8549, Sigma-Aldrich) was also included to prevent DNA origami structures from sticking to the walls of Eppendorf tubes[41]. After preparation the samples were thermally annealed as follows: samples were incubated at 90 °C for 5 min, cooled from 90 °C to 45 °C at 1 °C/min, held at 45 °C for 1 h, and then cooled from 45 °C to 20 °C at 0.1 °C/min. After annealing, seeds and caps were purified with a centrifugal filter (100 kDaA Amico Ultra-0.5 mL, Cat# UFC510096) to remove excess staple, adapter, and labeling strands. For purification, 50 μL of the annealed seed mixture and 250 μL of TAEM buffer were added to the filter and centrifuged at 2000 RCF for 4 min. The samples were washed three more times by adding 200 μL of TAEM buffer to the remaining solution and by repeating centrifugation; the last wash step was centrifuged at 3000 RCF. Purified seeds were stored at room temperature until used. Typically, seeds were annealed the day before they were used. Concentrations of the purified seeds were determined as previously described[38] (see Supplementary Note 14 for details).

**Nanotube growth**. For unregulated nanotube growth experiments, the five monomer strands for both the R and S monomers were mixed at equimolar concentrations in TAEM buffer with 0.05 mg/mL of biotinylated-BSA and 1 μM of a thymine 20-mer[37]. Samples were held at 90 °C for 5 min and then cooled to 20 °C at 1 °C/min. Purified seeds were added to the samples during the annealing process when the samples reached 30 °C. Since only 75% of seeds were viable for nucleation (Supplementary Note 6), the seeds were added to final concentrations of 0.01, 0.33, or 1 nM for the 0.075, 0.25, and 0.75 nM viable seed concentrations presented in the figures. After annealing the samples were incubated at 20 °C and aliquots were periodically taken for fluorescence imaging.

For buffer-regulated nanotube growth experiments, purified seeds, inactive monomers (5.5 μM), and $C_i$ strands (1.25 μM, unless otherwise stated) were mixed in TAEM buffer with 0.05 mg/mL of biotinylated-BSA and 1 μM of a thymine 20-mer. Pre-annealed $P_i$ complexes (5.5 μM) were added last to initiate the monomer buffering reactions. Samples were incubated at 20 °C and aliquots were periodically taken for fluorescence imaging.

**Fluorescence imaging and analysis**. Fluorescence imaging was conducted on an inverted microscope (Olympus IX71) using a 60×/1.45 NA oil immersion objective with 1.6x magnification. Images were captured on a cooled CCD camera (iXon3, Andor). For each imaging timepoint, a small aliquot (1/30th of the total reaction volume) was taken and diluted in TAEM with an additional 10 mM magnesium acetate for imaging, which facilitated nanotube binding to the glass coverslip. Samples with 0.075 nM seeds were typically diluted 100x for imaging, samples with 0.25 nM seeds were diluted 300x, and samples with 0.75 nM seeds were diluted 800x. After dilution, 5 μL of each sample was added to an 18 mm by 18 mm glass coverslip (Cat# 48366 045, VWR) that was then inverted onto a glass slide (Cat# 16004-424, VWR). Images were then captured at five to six randomly selected locations for each sample (corresponding to at least 50 seeds and nanotubes per sample at a given timepoint). The exact number of seeds and nanotubes analyzed for every timepoint of each sample are all presented in Supplementary Note 14. Images were processed and analyzed using custom MATLAB scripts (Supplementary Note 14). Consistent with previous studies[38], we found in experiments that a fraction of the origami seeds were unable to nucleate nanotube growth (Supplementary Note 6). We accounted for this effect in our experimental analysis and simulations and report viable seed concentrations throughout the text (Supplementary Note 6). Finally, nanotubes longer than 10–12 μm were prone to breaking during imaging (Supplementary Fig. 18), making it difficult to accurately quantify mean nanotube lengths for buffer-regulated growth with 0.075 nM seeds after 72 h. These timepoints were therefore omitted.

**Nanotube growth simulations**. We used a model of seeded nanotube growth consisting of two reversible reactions: monomer attachment and detachment (1) to seeds and (2) to nanotube growth faces. The kinetics of growth were modeled using exact sampling of trajectories of stochastic kinetics using the Gillespie algorithm[63] as in previous work[23,39]. The model had three parameters: the rate constant for monomer attachment to a seed and/or a growing nanotube face ($k_{ON}$), the rate of monomer detachment from a nanotube growth face ($k_{OFF, M-NT}$), and the rate of a monomer detachment from a seed ($k_{OFF, M-S}$). We used $k_{ON} = 2 \times 10^5$ M$^{-1}$ s$^{-1}$[39] and data from Fig. 2d, e to fit $k_{OFF, M-NT}$ and $k_{OFF, M-S}$ (Supplementary Note 4). Regulated growth was modeled by adding the monomer buffering reactions from Fig. 3 to the stochastic simulation with the reaction rate constants as presented in Fig. 7 (Supplementary Note 9).

**Reporting summary**. Further information on research design is available in the Nature Research Reporting Summary linked to this article.

## Data availability
The data for this study are available online at: https://doi.org/10.7281/T1/A9NKJ7.

## Code availability
The simulation code and image analysis code for this study are available at: https://doi.org/10.7281/T1/A9NKJ7.

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

## Acknowledgements

We thank Elisa Franco, Michael Pacella, and Sisi Jia for insightful conversations. This material is based upon work supported by the National Science Foundation Graduate Research Fellowship under Grant No. DGE-1232825. This work was supported by the Department of Energy under Grant No. DE-SC001 0426 for materials and supply costs.

## Author contributions

S.W.S, D.S., and R.S. designed the research. S.W.S conducted most of the experiments. T.M.M. performed the experiments presented in Fig. 6d–f. A.P. performed preliminary experiments for the study. S.W.S conducted the simulations. S.W.S and R.S. wrote the paper with feedback from the other authors.

## Competing interests

The authors declare no competing interests.
