## [Peer Review File · Nature Communications]

REVIEWER COMMENTS

Reviewer #1 (Remarks to the Author):

The paper from Schulman and coworkers describes a novel approach for controlled growth of DNA nanotubes crystals. The idea relies on the coupling of the crystallization process with a reversible reaction that replenishes the amount of monomer (i.e. setpoint concentration) necessary for sustained growth of the crystal in presence of seeds (i.e. load). This coupled reaction- inspired to the concept of chemical buffers- consists of a DNA "buffer", where the monomer in its inactive form (I) reacts with a DNA motifs (P) to produce the active monomer (M) and a consumer strand (C). The seeds are desired to maximize the degree of control over the crystallization process and improve the homogeneity of nanotubes lengths. The authors performed simulations to find out the parameters of the system (DNA sequences and concentrations values) that are necessary to achieve a regulated crystal growth in defined conditions and demonstrated the validity of their predictions by bench experiments. Observing the seeds and nanotubes by fluorescence microscopy at different time points, the authors characterized the kinetics of nanotubes growth in dependence of (i) the concentration of the seeds, (ii) the presence/absence of the monomer buffering reaction and (iii) the buffering capacity of this feedback reaction at different seeds concentrations, added either before or during the crystallization process. In all cases, the data show that the coupled feedback reaction works very well and can guarantee the formation of larger and more homogeneous crystals for a longer period of time. Finally, the authors discuss the small discrepancy observed between their simulations and experimental data and conclude that the crystallization process would not change appreciably even for reaction rates that are two orders of magnitude lower.

The conclusion is that the feedback reaction coupled to the crystallization process is extremely efficient and robust and that a similar concept could be applied to other crystals/buffer systems.

The work is extremely well-done and solid. The amount of data presented is massive and clearly denotes accurateness and commitment in the interpretation of the facts observed. I liked very much the analysis of the kinetic scheme with the Gillespie formalism and how the authors succeeded in implementing this method into their own system. The match between simulation and experimental data is convincing. I feel that the idea of coupling a reversible reaction to a process that consumes one component of this reaction is smart and can be used in other kinds of reactions to sustain their occurrence.

I am therefore glad to support the publication of this work in Nature Communications. Below are a few comments, which are either minor typos found on the manuscript or a few questions and remarks that the authors may consider to address before publication of the manuscript.

Comments/remarks:

In my opinion, Figure S8 in the SI gives a better overview of the system than Figure 3B in the main manuscript. The authors may consider to upgrade this figure?

page 10: "Although the seed concentration dictates the rate of monomer depletion, the depletion ratio is set by the concentrations of the buffering species and is thus the same irrespective of load". When looking at figure S8, how can this concept be expressed in terms of the rate constants involved in the entire process?

I have some problems in understanding figure 4D: whereas the black dashed line and the red line have a clear meaning, the green dots corresponding to the different seeds concentrations are less clear to me. There are few points for the low seeds conc in the unregulated growth and the green line on top seems to be an overlap of the three seeds conc?

page 10: "Buffer-regulated growth with $C_i = 1.25 \mu\text{M}$ resulted in more seeded nanotube growth than higher C_i concentrations (Supplementary Fig. S12) and less growth without seeds than $C_i = 1 \mu\text{M}$..." I could not find a reference figure to this statement. Can the authors indicate where the reader can find this comparison between $C=1.25 \mu\text{M}$ and $C= 1 \mu\text{M}$ in absence of seeds? In Fig. S14, the comparison is between $C= 1.25 \mu\text{M}$ and $C = 1.50 \mu\text{M}$ in seeded conditions and no unseeded growth is visible.

As a general comment, can the authors please indicate the number of nanotubes measured in the experiments (both main manuscript and SI)?

Also, the final section of the main manuscript could, in my opinion, be mostly transferred to the SI. This would not affect the quality of the paper and can make it a bit simpler to follow.

At this regard, I have a final question concerning the applicability of this buffering system. The authors show that the feedback reaction is effective even if the rate constants are both 2 orders of magnitude lower. Their ratio however keeps constant and equal to $K_{eq.} = 0.01$. How important is the ratio between the concentrations of the buffering species in making the buffer efficient? Is in general a factor 10x or 100x in respect to the depleted species a good choice?

Minor points:

page 1: "...notoriously difficult because crystallization proceeds depends ..." "proceeds" looks like a typo.

page 1: " ... measured AT supersaturation ..."

page 14: "... by lowering the effective BUT monomer attachment ..."

Reviewer #2 (Remarks to the Author):

The manuscript by Scaffter et al. reports the results of a highly novel investigation of self-regulated crystallization by means of a reversible buffering reaction that stabilizes the concentration of solute monomer as crystallization proceeds. Rather than becoming rapidly depleted to the solubility limit so that crystals are small and grow at rates that decrease over a significant range with time, or require initial high supersaturations that cause uncontrolled nucleation and high size dispersity, the authors show that the use of a reversible buffering reaction allows growth in a regime where seed crystals grow uniformly and at near constant rates to produce highly monodisperse crystals of a predictable size. They achieve this outcome for a system of DNA nanotubes through simulations used to model both monomer production and consumption and crystal nucleation and growth combined with experiments to test the model predictions. The experiments are extremely well designed and executed, the model is based on solid physical chemical principles, the presentation of the work is clear, the conclusions are well-supported by the results and the use of figures to illustrate the concepts and present the results are truly excellent. Personally, I learned a quite a lot from this manuscript and find it to be an important contribution to the field worthy of publication in Nature Communications with the following minor changes and corrections.

1) The use of the term "critical monomer concentration" is going to confuse many readers as the word "critical" in the context of crystallization usually refers to the concentration where nucleation occurs spontaneously — i.e., the end of the metastable zone. Strictly speaking there is nothing

wrong with the term, but it would be far better to call it the “equilibrium monomer concentration” or, equivalently, the solubility, because it is the concentration at which crystals no longer grow. If I were writing the paper, at its first use on page 2, I would write, “equilibrium monomer concentration or, equivalently, the solubility” and then use the word solubility for the rest of the manuscript.

2) The term “thermal annealing” is an odd choice for the intended purpose, as it usually refers to raising a system to a fixed temperature and letting it sit there for some period of time. The use of temperature to keep supersaturation constant is accomplished through continuous reduction of temperature (or continuous increase for a system exhibiting an inverse temperature dependence of the solubility). I suggest replacing thermal annealing” with “continuous decrease (or increase) of temperature”

3) The specific meaning of the term “capacity” in equation 3 is unclear to me. Assuming c is unitless, capacity has units of molarity. Is capacity then the total amount of monomer that can be produced before the monomer concentration drops more than a some given fraction of its starting value? How does one derive this?

4) In the caption to Figure 5, some text should be added to indicate that each of the plots in panel C refers to the experiments to the left (in panel A).

5) On page 12, the lack of any significant unseeded growth is used as a proxy to conclude that the introduction of caps to end growth also stops the production of active monomer. This seems unnecessarily indirect. Is there a reason one cannot directly measure the concentration rather than rely on the absence of an effect? Note that I do not think this is a critical measurement to make and consider it optional.

6) In the discussion, the difficulty of realizing closed-loop regulation of crystallization by titration is mentioned. In fact, for quite a number of systems, this is rather straightforward using an auto-titrator, which uses pH or an ion-specific electrode (or both) as the feedback signal to add titrants. This method is referred to as constant composition. It is used today by a number of groups to measure rates of nucleation and growth. George Nancollas was the leader in its application for many years. Others who use it today include Ruikang Tang and Helmut Coelfen. I have never seen it used as a means to keep growth rates constant, but it certainly could be used for that purpose.

7) I recommend that the authors take a look for literature where buffering reactions have been used or are believed to occur in nature. Two cases come to mind. One is calcium carbonate, for which I know there is an organic precursor molecule that decomposes to produce — I think — carbonate ions as calcium carbonate precipitates. I believe that Wolfgang Tremel’s group has used this. Then there is a natural source of phosphate that has been implicated in the production of calcium phosphate for bone formation. The two researchers I think of in this context are Sidney Omelon and Marc Grynpas.

8) Finally, there are a few typos I caught (new, corrected or text to be deleted in quotes):

- a. Page 2: crystals are used "to" facilitate
- b. Page 3: "similarly" robust growth
- c. Page 14: the effective "but" monomer attachment
- d. Page 16: self-assembly process "to" orchestrate

Reviewer #3 (Remarks to the Author):

Schaffter et al validate a system via simulation and experiment for buffering the concentration of free DNA tiles available for nanotube polymerization. This is a mechanism to maintain the

concentration of free monomers in a narrow range during the course of depletion of most of the total monomers. This could be highly useful for algorithmic assembly since for an unbuffered system, if free-monomer concentration is too high above the reversible concentration for growth, then a large amount of unwanted spurious nucleation and a large number of errors may result. Unlike annealing approaches, a buffering approach could adapt to varying rates of depletion of different tiles.

This is an outstanding contribution to the field. The manuscript is well written, the simulations and experiments performed were well done. Our team really enjoyed reading this manuscript.

We have the following minor suggestions.

We would appreciate discussion, either in the main text or the SI, of tradeoffs for buffering just one of the components, to obtain many of the benefits of buffering but requiring fewer extra strands.

Please specify sample sizes in figure legends.

Fig. 1

We suggest change of Fig. 1 legend from "new crystals nucleate continuously while other crystals grow" to "new crystals spuriously nucleate continuously while other crystals grow"

Fig. 1: We suggest coloring seeds red, including red seed in depiction of growing crystals as well.

Fig. 2

Please include discussion in the main text that added seed concentrations were 1 nM etc, and since ~75–80% of added seeds appeared to be viable, then this is why the viable seed concentrations were listed in figures as 0.75 nM etc.

It can be confusing at first to see the apparent fraction of viable seeds with nanotubes not hitting 100% in Figs. 5 and 6. We suggest authors enumerate more explicitly in SI section 5 why the apparent fraction of viable seeds with nanotubes are not plotted as 100% at the lowest (i.e. 0.075 nM) viable seed concentration (e.g. errors in estimating seed viability, variations in seed viability from prep to prep, sampling error in measurements). We also suggest that the authors explicitly mention in the main text that this discussion of why apparent fraction plotted differs from 100% can be found in SI section 5.

Fig. 3

In Fig. 3B, why are tiles depicted attaching to the seed with only one bond instead of two?

In Fig. 3D, we suggest flipping the orientation around the vertical axis for the lower left-hand producer complex, to match the orientations in the right-hand side of this figure section.

In Fig. 3D, we suggest including the lengths of domains as in Fig. S7B.

Fig. 4

In Fig. 4A unregulated growth, for 0.75 nM seed, why does simulated free monomer concentration dip below the equilibrium value before recovering?

For the Fig. 4A regulated growth panel, please mention in the figure caption that the initial increase in [Free Ms] is due to simulations starting with 0 free monomer and needing to equilibrate..

In the main text, the authors mention " the nanotubes from both the S1 and S2 seeds incorporated >50 nM more S monomers into nanotubes (nearly 23% more) after 48 hours (Fig.

6B) than nanotubes grown with the S1 seed concentration alone (Fig. 5B).” It looks to us that S2 seeded nanotubes are close to 4 μm in length at the 48 hour timepoint, while the S1 seeded nanotubes are close to 11 μm in length at the 48 hour timepoint. This would seem to correspond to $\sim 35\%$ increase in monomer incorporation.

Fig. 5

We suggest explicitly mentioning the seed concentrations for the three panels of Fig. 5C. We can see that they are supposed to line up with 5A, but this might not be completely clear to the reader.

The histograms presented in panel C and elsewhere in the paper are a little hard to read. Assuming the sample sizes are large enough, it might be clearer to plot kernel density estimates and then providing the raw histogram data in the supplement.

We suggest replotting the data from Fig. 5B in Fig. S11 to make it easier to compare when looking at S11.

Fig. 6

For Fig. 6D, would it be possible to denote the caps using a different marker/color to that used for S2 seeds in Fig. 6A? This is clear from the caption, but the figures would be easier to visually understand if the markers were different.

For Fig. 6F, why is there an apparent burst of spurious nucleation such that the ratio between unseeded and seeded is close to 20%? Is this resulting from unseeded nucleation, or breakage of tubes from their seeds in preparation of the samples? Please clarify in the text/caption.

Fig. 7

For panels C–D and the equilibrium line, please add more data points so that the thin line isn’t so jagged. It was a little distracting in the current form.

We suggest adding a sentence describing why equilibrium active monomer concentrations are depleted more slowly for slow buffering (i.e. due to slower growth)

Add to clarify what is meant

On page 8, first paragraph, second sentence, it would be nice to add the variable I_s in parentheses: “To obtain a high capacity, we selected concentrations of the inactive monomers (I_s) and P_i to both be 5.5 μM (>35 times the setpoint concentration).”

Typos

SI, page 31, third paragraph, fourth sentence: “However, if the rate constants for the monomer buffering reactions were much lower, we might expect to see slower nanotube growth rates as growth would be deplete monomers faster than monomer buffering could replenish them.” We believe the word “be” should be removed.

Page 14, first paragraph, third sentence: “Competition with buffering species to bind to growth sites might also influence the growth rate by lowering the effective but monomer attachment rate constant (k_{on}), but simulations with reduced k_{on} values did not recapitulate the different initial growth rates we observed across seed concentrations (Supplementary Fig. S17).” We believe the word “but” should be removed.

Page 12 paragraph 3 “monomer concentration still be high enough” should be “monomer concentration was still high enough”

Page 18 nanotube growth simulations “Gillespie algorithm as in previously work” should be “Gillespie algorithm as in previous work”

Supplementary section 1.2 page 3 “staple strand sequences are the same as those used in a previously work” should be “staple strand sequences are the same as those used in previous work”

Figure S2 panel D – the caption isn't clear (it's referring to dashed/solid lines but the figure has histograms)

Supplementary section 8.2 page 22 "over the course of the simulations are virtually identical the concentrations" should be "over the course of the simulations are virtually identical TO the concentrations"

Supplementary section page 31 last paragraph: k_f and k_r values are repeated, one of those should be 10^1 and 10^3 instead of 10^0 and 10^2

Reviewer #1 (Remarks to the Author):

The paper from Schulman and coworkers describes a novel approach for controlled growth of DNA nanotubes crystals. The idea relies on the coupling of the crystallization process with a reversible reaction that replenishes the amount of monomer (i.e. setpoint concentration) necessary for sustained growth of the crystal in presence of seeds (i.e. load). This coupled reaction- inspired to the concept of chemical buffers- consists of a DNA “buffer”, where the monomer in its inactive form (I) reacts with a DNA motifs (P) to produce the active monomer (M) and a consumer strand (C). The seeds are desired to maximize the degree of control over the crystallization process and improve the homogeneity of nanotubes lengths. The authors performed simulations to find out the parameters of the system (DNA sequences and concentrations values) that are necessary to achieve a regulated crystal growth in defined conditions and demonstrated the validity of their predictions by bench experiments.

Observing the seeds and nanotubes by fluorescence microscopy at different time points, the authors characterized the kinetics of nanotubes growth in dependence of (i) the concentration of the seeds, (ii) the presence/absence of the monomer buffering reaction and (iii) the buffering capacity of this feedback reaction at different seeds concentrations, added either before or during the crystallization process. In all cases, the data show that the coupled feedback reaction works very well and can guarantee the formation of larger and more homogeneous crystals for a longer period of time. Finally, the authors discuss the small discrepancy observed between their simulations and experimental data and conclude that the crystallization process would not change appreciably even for reaction rates that are two orders of magnitude lower.

The conclusion is that the feedback reaction coupled to the crystallization process is extremely efficient and robust and that a similar concept could be applied to other crystals/buffer systems.

The work is extremely well-done and solid. The amount of data presented is massive and clearly denotes accurateness and commitment in the interpretation of the facts observed. I liked very much the analysis of the kinetic scheme with the Gillespie formalism and how the authors succeeded in implementing this method into their own system. The match between simulation and experimental data is convincing. I feel that the idea of coupling a reversible reaction to a process that consumes one component of this reaction is smart and can be used in other kinds of reactions to sustain their occurrence.

I am therefore glad to support the publication of this work in Nature Communications.

Below are a few comments, which are either minor typos found on the manuscript or a few questions and remarks that the authors may consider to address before publication of the manuscript.

Comments/remarks:

In my opinion, Figure S8 in the SI gives a better overview of the system than Figure 3B in the main manuscript. The authors may consider to upgrade this figure?

We appreciate the reviewer’s suggestion for improved clarity. We have modified the schematic in Figure 3B to more closely resemble the schematic in Figure S8.

page 10: “Although the seed concentration dictates the rate of monomer depletion, the depletion ratio is set by the concentrations of the buffering species and is thus the same irrespective of load”. When looking at figure S8, how can this concept be expressed in terms of the rate constants involved in the entire process?

We agree with the reviewer that our discussion of this concept could be improved. We have expanded this discussion as follows in the revised manuscript: “Although the seed concentration (*i.e.* the load) dictates the rate at which monomers are depleted (and the total amount of monomers depleted during a given growth time), the depletion ratio, *i.e.* the amount the setpoint concentration drops with respect to the concentration of monomers incorporated during growth, is the same irrespective of load (Fig. 4d). The depletion ratio is analogous to buffer capacity and is thus set by the initial concentration of the buffering species (Eq. 3).”

The exact relationship for how buffer capacity coefficient (c - in Eq. 3) depends on rate constants and concentration ratios is quite complex (derived in Ref (1)) so we do not believe that including that expression in the text would add any clarity. We have explicitly added a statement referring the reader to Ref for the derivation of Eq. 3 in the revised manuscript.

I have a problem understanding figure 4D: whereas the black dashed line and the red line have a clear meaning, the green dots corresponding to the different seeds concentrations are less clear to me. There are few points for the low seeds conc in the unregulated growth and the green line on top seems to be an overlap of the three seeds conc?

We appreciate the reviewer bringing this up and giving us the opportunity to clarify our discussion here. We have added the following clarification to the caption of Figure 4: “The green dashed lines show the change in M_S vs. total monomer incorporation during buffer regulated growth. These lines have the same slope for all seed concentrations (and therefore overlap) because the amount the setpoint drops per M_S incorporated is a constant irrespective of load. The slope of these overlapping lines is the depletion ratio. The red line indicates the critical monomer concentration for growth. When the change in free M_S reaches this line, growth will stop. The higher the seed concentration, the faster this line is reached and the buffer is exhausted”.

page 10: “Buffer-regulated growth with $C_i = 1.25 \mu\text{M}$ resulted in more seeded nanotube growth than higher C_i concentrations (Supplementary Fig. S12) and less growth without seeds than $C_i = 1 \mu\text{M}$...” I could not find a reference figure to this statement. Can the authors indicate where the reader can find this comparison between $C=1.25 \text{ uM}$ and $C= 1 \text{ uM}$ in absence of seeds? In Fig. S14, the comparison is between $C= 1.25 \text{ uM}$ and $C = 1.50 \text{ uM}$ in seeded conditions and no unseeded growth is visible.

This is a good point. This comparison was provided in the SI but we did not point the reader to that comparison in the main text. Supplementary Figure S13 shows the growth in the absence of seeds for all C_i concentrations tested (including $C_i = 1 \text{ uM}$, $C_i = 1.25 \text{ uM}$ and unregulated growth with 150 nM monomers), allowing the reader to compare the amount of growth by observing the number of nanotubes in the micrographs corresponding to growth at different conditions. The way we referred the reader to this figure was confusing as we referenced multiple different figures. To improve clarity we have updated this to: “...and less growth without seeds than

$C_i = 1 \mu\text{M}$, with unseeded growth comparable to unregulated growth with 150 nM monomers (Supplementary Figs. S13)”.

As a general comment, can the authors please indicate the number of nanotubes measured in the experiments (both main manuscript and SI)?

We appreciate the reviewer bringing this to attention. As the exact number of nanotubes and seeds analyzed varies across each sample and timepoint, providing each sample sizes in lists would make the captions clumsy to read. But we have now included this information – we have tabulated all of these sample sizes in Supplementary Section 14. In each relevant figure caption, we have added a statement that directs the reader to Supplementary Section 14 to find the exact sample sizes for each sample and timepoint. We have also added this statement to the methods as well. In all cases except for two from Figure 6b, $N > 70$ and in almost all $N > 100$.

Also, the final section of the main manuscript could, in my opinion, be mostly transferred to the SI. This would not affect the quality of the paper and can make it a bit simpler to follow.

We appreciate the reviewer’s thoughts here. We also respect the fact that this paper contains a range of results and ideas that may not all be interpreted in a single read. However, we do think this section is important for understanding the buffering process and the parameters that buffering is sensitive / insensitive to. Along these lines, per the reviewer’s comment below and comments from reviewer 3, we have modified this section a bit to focus on the important design considerations for obtaining effective buffering and the interesting tradeoffs that can emerge when choosing the parameters and concentrations for the reaction.

At this regard, I have a final question concerning the applicability of this buffering system. The authors show that the feedback reaction is effective even if the rate constants are both 2 orders of magnitude lower. Their ratio however keeps constant and equal to $K_{eq} = 0.01$. How important is the ratio between the concentrations of the buffering species in making the buffer efficient? Is in general a factor 10x or 100x in respect to the depleted species a good choice?

The reviewer brings up a great point here that is an important consideration for designing a buffering scheme for other crystallization processes. The K_{eq} for a given buffering reaction will dictate the exact ratio of buffering species required to obtain a desired setpoint (Eq. 2 in the main text) and this equilibrium constant may be set by the specific chemistry of the process to be buffered. However, as long as the appropriate concentration ratio is maintained, in principle, any concentration of buffering species could be used. The capacity of the buffer (*i.e.* how much monomer it can produce before the setpoint decreases by a specified percentage) increases with increasing buffering species concentrations while maintaining the equilibrium constant. So, the higher the concentration of buffering species relative to the setpoint concentration, the higher the capacity (Eq. 3 of the main text). In our system we used an inactive monomer concentration roughly 36 times that of the desired active monomer concentration and were able to obtain greater than 18-fold more monomers incorporated into nanotubes than with unregulated growth. In Supp. Section 12, using simulations we demonstrate how the capacity increases when inactive monomer concentrations are increased to 73, 183, and 360 times greater than the setpoint active monomer concentrations. However, we found that in experiments there are side reactions

between the buffering species and the active monomers / monomer attachment sites on nanotubes that can impede growth. The higher the concentration of buffering species, the more prevalent these side reactions. In Supp. Section 13, we conduct some analysis that suggests that using buffering species concentrations greater than 100 times the setpoint active monomer concentration could significantly impede crystallization. Thus, for our specific system we believe the buffering species should be greater than 10 times and less than 100 times the desired setpoint monomer concentration to ensure enhanced growth with minimal negative effects from side reactions.

We have updated the last paragraph of page 16 in the main text to illustrate these design considerations and trade-offs.

page 1: “..notoriously difficult because crystallization proceeds depends ...” “proceeds” looks like a typo.

The reviewer is correct and we have removed the word “proceeds” here.

page 1: “ ... measured AT supersaturation ...”

This wasn't a typo but we see that this wording was misleading. We intended to say that the difference the monomer concentration and the critical concentration is defined as supersaturation. We have changed this to statement to “defined as supersaturation”

page 14: “.... by lowering the effective BUT monomer attachment ...”

The reviewer is correct and we have removed the word “but” here.

Reviewer #2 (Remarks to the Author):

The manuscript by Schaffter et al. reports the results of a highly novel investigation of self-regulated crystallization by means of a reversible buffering reaction that stabilizes the concentration of solute monomer as crystallization proceeds. Rather than becoming rapidly depleted to the solubility limit so that crystals are small and grow at rates that decrease over a significant range with time, or require initial high supersaturations that cause uncontrolled nucleation and high size dispersity, the authors show that the use of a reversible buffering reaction allows growth in a regime where seed crystals grow uniformly and at near constant rates to produce highly monodisperse crystals of a predictable size. They achieve this outcome for a system of DNA nanotubes through simulations used to model both monomer production and consumption and crystal nucleation and growth combined with experiments to test the model predictions. The experiments are extremely well designed and executed, the model is based on solid physical chemical principles, the presentation of the work is clear, the conclusions are well-supported by the results and the use of figures to illustrate the concepts and present the results are truly excellent. Personally, I learned quite a lot from this manuscript and find it to be an important contribution to the field worthy of publication in Nature Communications with the following minor changes and corrections.

1) The use of the term “critical monomer concentration” is going to confuse many readers as the word “critical” in the context of crystallization usually refers to the concentration where nucleation occurs spontaneously — i.e., the end of the metastable zone. Strictly speaking there is

nothing wrong with the term, but it would be far better to call it the “equilibrium monomer concentration” or, equivalently, the solubility, because it is the concentration at which crystals no longer grow. If I were writing the paper, at its first use on page 2, I would write, “equilibrium monomer concentration or, equivalently, the solubility” and then use the word solubility for the rest of the manuscript.

We appreciate the reviewer for bringing this potentially confusing nomenclature to our attention and we realize that we did not explicitly define what critical monomer concentration meant when we first use the term in the paper. By critical monomer concentration we are referring to the free monomer concentration at which the rate of monomer attachment and detachment are equal *i.e.* the concentration where no net crystal growth will occur. We recognize that this term may mean something slightly different in other fields but it is typically used as we have defined it not just here and in other related work on DNA crystallization (Refs (2)) but also in work on actin and microtubule crystallization (Refs (3–5)). To avoid confusion, we have modified the text such that we define critical monomer concentration explicitly when we introduce it: “...the critical monomer concentration – the free monomer concentration at which no net crystal growth occurs...”.

2) The term “thermal annealing” is an odd choice for the intended purpose, as it usually refers to raising a system to a fixed temperature and letting it sit there for some period of time. The use of temperature to keep supersaturation constant is accomplished through continuous reduction of temperature (or continuous increase for a system exhibiting an inverse temperature dependence of the solubility). I suggest replacing “thermal annealing” with “continuous decrease (or increase) of temperature”

We thank the reviewer for pointing out this confusion. We have updated this accordingly in the main text. We agree with the reviewer that our use of thermal annealing was not specifically defined. We still use this term in the methods section to describe how we prepared the DNA origami seeds and caps and the buffering species as this is the standard term used for the preparation of these DNA components (Refs ((6–8))). These DNA components were prepared by heating the samples to 90 °C for 5 minutes to ensure all the strands present were not hybridized at all with themselves or other components. After this initial heating step, we then slowly cooled the samples to 20 °C to ensure they assembled into the thermodynamically favored final structures. We have updated the methods section to explicitly define that this is what we mean by thermal annealing for our sample preparation.

3) The specific meaning of the term “capacity” in equation 3 is unclear to me. Assuming c_- is unitless, capacity has units of molarity. Is capacity then the total amount of monomer that can be produced before the monomer concentration drops more than a some given fraction of its starting value? How does one derive this?

The reviewer’s definition of capacity is correct and capacity as we used it has units of molarity. c_- , the capacity coefficient, is unitless. We have reworded this section to improve clarity. Further, as c_- and buffer capacity have been previously derived, we added a statement directing the reader to the previous work for the derivation: Ref (1).

4) In the caption to Figure 5, some text should be added to indicate that each of the plots in panel C refers to the experiments to the left (in panel A).

Thank you for this suggestion. We have added this statement for clarity.

5) On page 12, the lack of any significant unseeded growth is used as a proxy to conclude that the introduction of caps to end growth also stops the production of active monomer. This seems unnecessarily indirect. Is there a reason one cannot directly measure the concentration rather than rely on the absence of an effect? Note that I do not think this is a critical measurement to make and consider it optional.

The reviewer brings up great point here. In principle, one could track the rate of monomer production with fluorescence since only the active monomers exhibit unquenched fluorescence. So as nanotube growth proceeds, the fluorescence signal should continually increase. We initially designed our monomer labeling scheme this way exactly with the goal of directly measuring this change in monomer concentration. However, we ran into technical challenges when trying to use this approach with our equipment. We studied nanotube growth using monomer concentrations such that heterogeneous nucleation of nanotubes from seeds was favorable at a temperature of 20C but homogeneous nucleation of nanotubes was not. Unfortunately, we did not have an instrument that could both hold the temperature of the samples at 20C and track fluorescence at the same time. And because 20C is slightly cooler than room temperature, we were not successful in performing these measurements without temperature control because fluctuations in lab temperature (up to 25C) resulted in inconsistent nanotube growth across experiments. Given measuring nanotube length still provided convincing, although indirect, evidence for capping preventing further active monomer production, we chose not to further pursue the direct measurement of monomer concentrations.

6) In the discussion, the difficulty of realizing closed-loop regulation of crystallization by titration is mentioned. In fact, for quite a number of systems, this is rather straightforward using an auto-titrator, which uses pH or an ion-specific electrode (or both) as the feedback signal to add titrants. This method is referred to as constant composition. It is used today by a number of groups to measure rates of nucleation and growth. George Nancollas was the leader in its application for many years. Others who use it today include Ruikang Tang and Helmut Coelfen. I have never seen it used as a means to keep growth rates constant, but it certainly could be used for that purpose.

We appreciated this suggestion as we were not familiar with the constant composition technique. We were considering systems for biomolecular crystallization when we formulated our argument, where generally a simple measurement like pH is not sufficient to measure the state of supersaturation. Given this reviewer's point, we agree that our discussion of closed-loop crystallization could be corrected and improved. We have removed the statement that titration cannot be used for closed loop control in our revised manuscript. We have also rewritten this portion of the discussion (page 17, paragraph 3) to discuss how continuous flow reactors and auto-titration can

be used as a physical means of control while we have introduced a chemical means of achieving similar control over crystallization.

7) I recommend that the authors take a look for literature where buffering reactions have been used or are believed to occur in nature. Two cases come to mind. One is calcium carbonate, for which I know there is an organic precursor molecule that decomposes to produce — I think — carbonate ions as calcium carbonate precipitates. I believe that Wolfgang Tremel's group has used this. Then there is a natural source of phosphate that has been implicated in the production of calcium phosphate for bone formation. The two researchers I think of in this context are Sidney Omelon and Marc Grynepas.

We appreciate the reviewer's suggestions here. We learned a lot from them and agree these points are highly relevant to our work and should be discussed in our paper. We have added some references to the work of Sidney Omelon and Marc Grynepas to the introduction at the point where we discuss chemical regulation of crystallization in biology. We have also added a discussion of Wolfgang Tremel's work on converting an organic precursor into carbonate for calcium carbonate crystallization to the discussion (page 17, paragraph 2); specifically we discuss how the existence of such inactive precursors could facilitate the adoption of monomer buffering for many other crystallization processes.

8) Finally, there are a few typos I caught (new, corrected or text to be deleted in quotes):

a. Page 2: crystals are used "to" facilitate

The reviewer is correct and we have added the word "to" here.

b. Page 3: "similarly" robust growth

The reviewer is correct and we have changed the word "similar" to "similarly" here.

c. Page 14: the effective "but" monomer attachment

The reviewer is correct and we have removed the word "but" here.

d. Page 16: self-assembly process "to" orchestrate

The reviewer is correct that this sentence was grammatically incorrect and we have corrected it.

Reviewer #3 (Remarks to the Author):

Schaffter et al validate a system via simulation and experiment for buffering the concentration of free DNA tiles available for nanotube polymerization. This is a mechanism to maintain the concentration of free monomers in a narrow range during the course of depletion of most of the total monomers. This could be highly useful for algorithmic assembly since for an unbuffered system, if free-monomer concentration is too high above the reversible concentration for growth, then a large amount of unwanted spurious nucleation and a large number of errors may result. Unlike annealing approaches, a buffering approach could adapt to varying rates of depletion of different tiles.

This is an outstanding contribution to the field. The manuscript is well written, the simulations and experiments performed were well done. Our team really enjoyed reading this manuscript.

We appreciate the reviewer's kind remarks about this work.

We have the following minor suggestions.

We would appreciate discussion, either in the main text or the SI, of tradeoffs for buffering just one of the components, to obtain many of the benefits of buffering but requiring fewer extra strands.

The reviewer brings up a great point here. It should be possible to merely buffer one of the two monomer types in our system and supply the other monomer type at a high concentration and still achieve the same regulated growth results. Indeed, this is an advantage that would be very useful to employ in more complex multi-component self-assembly processes to simplify the buffering scheme. We have added a brief discussion of this possibility at the end of the first paragraph on page 16 of the main text.

Please specify sample sizes in figure legends.

We appreciate the reviewer bringing this to attention. As the exact number of nanotubes and seeds analyzed varies across each sample and timepoint, providing each sample sizes in lists would make the captions clumsy to read. But we have now included this information – we have tabulated all of these sample sizes in Supplementary Section 14. In each relevant figure caption, we have added a statement that directs the reader to Supplementary Section 14 to find the exact sample sizes for each sample and timepoint. We have also added this statement to the methods as well. In all cases except for two from Figure 6b, $N > 70$ and in almost all $N > 100$.

Fig. 1

We suggest change of Fig. 1 legend from “new crystals nucleate continuously while other crystals grow” to “new crystals spuriously nucleate continuously while other crystals grow”

We appreciate the reviewer's suggestion here. We have changed the statement to “new crystals homogeneously nucleate (i.e. not from seeds) continuously as other crystals grow”.

Fig. 1: We suggest coloring seeds red, including red seed in depiction of growing crystals as well.

We have updated Figure 1 to depict the seeds in red. We also added a clarifying statement to the figure caption to inform the reader that the crystals in the schematics are 3D crystals.

Fig. 2

Please include discussion in the main text that added seed concentrations were 1 nM etc, and since ~75–80% of added seeds appeared to be viable, then this is why the viable seed concentrations were listed in figures as 0.75 nM etc.

We agree with the reviewer that this is an important point that should be included in the text. We have added a statement discussing this in the main text when we first introduce the nanotube growth experiments (page 6, paragraph 2). Additionally, we have added a statement to the “Nanotube growth experiments” section of the methods that highlights that seeds were added to 0.01, 0.33, and 1 nM to obtain the viable seed concentrations of 0.075, 0.25, and 0.75 nM, respectively.

It can be confusing at first to see the apparent fraction of viable seeds with nanotubes not hitting 100% in Figs. 5 and 6. We suggest authors enumerate more explicitly in SI section 5 why the apparent fraction of viable seeds with nanotubes are not plotted as 100% at the lowest (i.e. 0.075 nM) viable seed concentration (e.g. errors in estimating seed viability, variations in seed viability from prep to prep, sampling error in measurements). We also suggest that the authors explicitly mention in the main text that this discussion of why apparent fraction plotted differs from 100% can be found in SI section 5.

We appreciate the reviewer bringing up this important point. We assumed that 75% of the seeds were viable for all our experiments, however, as the reviewer points out this value varied somewhat from experiment to experiment as well as across timepoints due to sample/measurement errors (each timepoint is an independently prepared slide that is imaged). Thus, we observed between 70-80% of seeds nucleated nanotubes depending on the experiment and or timepoint. We opted to use a global value of 75% for the viable seed correction as this resulted in fractions of viable seeds with nanotubes very close to 1 most of the timepoints for our regulated growth experiments while also ensuring that these fractions did not exceed 1 for most of our results.

We have added this discussion to Supplementary Section 5 to clarify this point to the reader. Additionally, we have added a statement to the caption for Figure 5D that explains this experimental variation and directs the reader to Supplementary Section 5 to look at the fraction of viable seeds with nanotubes across all timepoints – most of which are very close to 1 for regulated growth.

Fig. 3

In Fig. 3B, why are tiles depicted attaching to the seed with only one bond instead of two?

We appreciate the reviewer point this out - there was an error in the drawing. We have updated this to show monomers attached to the nanotube growth face with two bonds.

In Fig. 3D, we suggest flipping the orientation around the vertical axis for the lower left-hand producer complex, to match the orientations in the right-hand side of this figure section.

We appreciate the reviewer’s suggestion here, however, we oriented the Producer complex such that the e domain on the complex aligns with the e’ domain on the inactive monomer. This thus aligns the toehold domains and the complexes in the orientation for which they will undergo strand displacement.

In Fig. 3D, we suggest including the lengths of domains as in Fig. S7B.

We have updated the figure to include the domain lengths.

Fig. 4

In Fig. 4A unregulated growth, for 0.75 nM seed, why does simulated free monomer concentration dip below the equilibrium value before recovering?

We appreciate the reviewer pointing this out. We realize that this was a numerical issue arising in our simulation. In the Gillespie algorithm only a specified number of molecules are tracked. In most our simulations we tracked 250 seeds. However, the higher the seed concentration the smaller the simulation volume and thus the fewer monomers tracked. Thus, at the 0.75 nM seed concentration a slight fluctuation in the free monomer number in the simulation scales to an appreciable concentration fluctuation. We have rerun this simulation tracking 750 seeds for the 0.75 nM seed sample which removed this issue and we have updated Figure 4A accordingly.

For the Fig. 4A regulated growth panel, please mention in the figure caption that the initial increase in [Free Ms] is due to simulations starting with 0 free monomer and needing to equilibrate.

We have added this statement to the figure caption.

In the main text, the authors mention “ the nanotubes from both the S1 and S2 seeds incorporated >50 nM more S monomers into nanotubes (nearly 23% more) after 48 hours (Fig. 6B) than nanotubes grown with the S1 seed concentration alone (Fig. 5B).” It looks to us that S2 seeded nanotubes are close to 4 μm in length at the 48 hour timepoint, while the S1 seeded nanotubes are close to 11 μm in length at the 48 hour timepoint. This would seem to correspond to ~35% increase in monomer incorporation.

We appreciate the reviewer pointing this out. Since we do not supply the numbers involved in this calculation in the text this calculation could be confusing to the reader.

Here we are comparing the total length of the S1 and S2 nanotubes in Figure 6B to the length of the 0.075 nM seed experiment in Figure 5B after 48 hours. The exact numbers are:

For Figure 6B at 48 hours: S1 nanotubes = 10.49 μm , S2 nanotubes = 3.29 μm , combined length = 13.78 μm

For Figure 5B at 48 hours for the 0.075 nM seeds: Nanotubes = 11.1 μm

Increase = $13.78/11.1 = 1.24$ or 24% increase.

To clarify this for the reader we have updated the text to include the numbers involved in this calculation: “based on mean lengths, the nanotubes from both the S1 and S2 seeds (combined mean length of 13.78 μm) incorporated >50 nM more S monomers into nanotubes (24% more) after 48 hours (Fig. 6B) than nanotubes grown with the 0.075 nM seed concentration alone (mean length of 11.1 μm) (Fig. 5B).”

Fig. 5

We suggest explicitly mentioning the seed concentrations for the three panels of Fig. 5C. We can

see that they are supposed to line up with 5A, but this might not be completely clear to the reader.

We have added these concentrations to the panels in Fig. 5C.

The histograms presented in panel C and elsewhere in the paper are a little hard to read. Assuming the sample sizes are large enough, it might be clearer to plot kernel density estimates and then providing the raw histogram data in the supplement.

We appreciate the reviewer's suggestion about using kernel density estimation as a way to smooth the graphs. This is a very useful way to improve clarity, but we do not think that we have large enough sample sizes to present our results in this way. The graphs in Figure 5C, for example, have 25 bars but less than 200 data points in each histogram. Smoothing such histograms with KDE would introduce significant error which would need to be diagrammed – we fear that the resulting clutter would obviate the benefits of using smoothed curves.

We suggest replotting the data from Fig. 5B in Fig. S11 to make it easier to compare when looking at S11.

We agree with the reviewer's suggestion here. We have updated Fig. S11 to include the data from Fig. 5B, D, and E for easier comparison.

Fig. 6

For Fig. 6D, would it be possible to denote the caps using a different marker/color to that used for S2 seeds in Fig. 6A? This is clear from the caption, but the figures would be easier to visually understand if the markers were different.

We appreciate the reviewer bringing this potentially confusing point to our attention. We have changed the color of the caps to yellow to differentiate them from the S2 seeds.

For Fig. 6F, why is there an apparent burst of spurious nucleation such that the ratio between unseeded and seeded is close to 20%? Is this resulting from unseeded nucleation, or breakage of tubes from their seeds in preparation of the samples? Please clarify in the text/caption.

We believe this is primarily due to some unavoidable spurious nucleation. We found that in most experiments, both regulated and unregulated with 150 nM monomers, at least 5-10% of nanotubes would be unseeded. This background (spurious) nucleation varied somewhat from experiment to experiment ranging as high as 15-20% in some experiments. We have added a statement to the caption of Figure 6 specifying this.

Fig. 7

For panels C–D and the equilibrium line, please add more data points so that the thin line isn't so jagged. It was a little distracting in the current form.

We have added more data points to the equilibrium lines as suggested.

We suggest adding a sentence describing why equilibrium active monomer concentrations are depleted more slowly for slow buffering (i.e. due to slower growth)

We have added a sentence about this point on page 14, paragraph 2 .

Add to clarify what is meant

On page 8, first paragraph, second sentence, it would be nice to add the variable I_s in parentheses: “To obtain a high capacity, we selected concentrations of the inactive monomers (I_s) and P_i to both be $5.5 \mu\text{M}$ (>35 times the setpoint concentration).”

We thank the reviewer for pointing this out to enhance the clarity of the manuscript. We have updated this instance as well as other instances in the text where inactive monomers are referenced.

Typos

SI, page 31, third paragraph, fourth sentence: “However, if the rate constants for the monomer buffering reactions were much lower, we might expect to see slower nanotube growth rates as growth would be deplete monomers faster than monomer buffering could replenish them.” We believe the word “be” should be removed.

The reviewer is correct and we have removed the word “be” here.

Page 14, first paragraph, third sentence: “Competition with buffering species to bind to growth sites might also influence the growth rate by lowering the effective but monomer attachment rate constant (k_{on}), but simulations with reduced k_{on} values did not recapitulate the different initial growth rates we observed across seed concentrations (Supplementary Fig. S17).” We believe the word “but” should be removed.

The reviewer is correct and we have removed the word “but” here.

Page 12 paragraph 3 “monomer concentration still be high enough” should be “monomer concentration was still high enough”

The reviewer is correct and we have corrected this mistake.

Page 18 nanotube growth simulations “Gillespie algorithm as in previously work” should be “Gillespie algorithm as in previous work”

The reviewer is correct and we have changed “previously” to “previous”.

Supplementary section 1.2 page 3 “staple strand sequences are the same as those used in a previously work” should be “staple strand sequences are the same as those used in previous work”

The reviewer is correct and we have changed “previously” to “previous”.

Figure S2 panel D – the caption isn’t clear (it’s referring to dashed/solid lines but the figure has histograms)

We appreciate the reviewer pointing out this confusing point. We have updated the figure caption to state that in panels A-C the dashed/solid lines are simulation/experimental data, respectively.

Supplementary section 8.2 page 22 “over the course of the simulations are virtually identical the concentrations” should be “over the course of the simulations are virtually identical TO the concentrations”

The reviewer is correct and we have added the word “to” here.

Supplementary section page 31 last paragraph: k_f and k_r values are repeated, one of those should be 10^1 and 10^3 instead of 10^0 and 10^2

The reviewer is correct and we corrected this mistake.

References

1. D. Scalise, N. Dutta, R. Schulman, DNA Strand Buffers. *J. Am. Chem. Soc.* **140**, 12069–12076 (2018).
2. R. F. Hariadi, B. Yurke, E. Winfree, Thermodynamics and kinetics of DNA nanotube polymerization from single-filament measurements. *Chem. Sci.* **6**, 2252–2267 (2015).
3. L. Blanchoin, R. Boujemaa-Paterski, C. Sykes, J. Plastino, Actin Dynamics, Architecture, and Mechanics in Cell Motility. *Physiol. Rev.* **94**, 235–263 (2014).
4. S. Petry, R. D. Vale, Microtubule nucleation at the centrosome and beyond. *Nat. Cell Biol.* **17**, 1089–1093 (2015).
5. A. Wegner, G. Isenberg, 12-fold difference between the critical monomer concentrations of the two ends of actin filaments in physiological salt conditions. *Proc Natl Acad Sci USA* **80**, 4922 (1983).
6. X. Wei, J. Nangreave, S. Jiang, H. Yan, Y. Liu, Mapping the Thermal Behavior of DNA Origami Nanostructures. *J. Am. Chem. Soc.* **135**, 6165–6176 (2013).
7. C. E. Castro, *et al.*, A primer to scaffolded DNA origami. *Nature Methods* **8**, 221–229 (2011).
8. C. G. Evans, E. Winfree, Physical principles for DNA tile self-assembly. *Chem. Soc. Rev.* **46**, 3808–3829 (2017).

REVIEWERS' COMMENTS

Reviewer #1 (Remarks to the Author):

The authors answered all points in a spectacular fashion.
Great work.

Reviewer #2 unfortunately did not respond to our invitation to review.

Reviewer #3 (Remarks to the Author):

I am satisfied with all author responses and recommend publication with no further modifications.